# Fatal Lassa fever in cynomolgus monkeys is associated with systemic viral dissemination and inflammation

Jimmy Hortion[1,2], Emeline Perthame[3], Blaise Lafoux[1,2], Laura Soyer[1,2], Stéphanie Reynard[1,2], Alexandra Journeaux[1,2], Clara Germain[1,2], Hélène Lopez-Maestre[3], Natalia Pietrosemoli[3], Nicolas Baillet[1,2], Séverine Croze[4], Catherine Rey[4], Catherine Legras-Lachuer[4], Sylvain Baize[1,2]*

1 Unité de Biologie des Infections Virales Emergentes, Institut Pasteur, Université Paris Cité, Lyon, France, 2 Centre International de Recherche en Infectiologie (CIRI), Université de Lyon, INSERM U1111, Ecole Normale Supérieure de Lyon, Université Lyon 1, CNRS UMR5308, Lyon France, 3 Institut Pasteur, Université Paris Cité, Bioinformatics and Biostatistics Hub, Paris, France, 4 ViroScan3D SAS, Trevoux, France

* sylvain.baize@pasteur.fr

## Abstract

The pathogenesis of Lassa fever has not yet been fully deciphered, particularly as concerns the mechanisms determining whether acute infection is controlled or leads to catastrophic illness and death. Using a cynomolgus monkey model of Lassa virus (LASV) infection reproducing the different outcomes of the disease, we performed histological and transcriptomic studies to investigate the dynamics of LASV infection and the immune mechanisms associated with survival or death. Lymphoid organs are an early major reservoir for replicating virus during Lassa fever, with LASV entering through the cortical sinus of draining lymph nodes regardless of disease outcome. However, subsequent viral tropism varies considerably with disease severity, with viral dissemination limited almost entirely to lymphoid organs and immune cells during nonfatal Lassa fever. By contrast, the systemic dissemination of LASV to all organs and diverse cell types, leading to infiltrations with macrophages and neutrophils and an excessive inflammatory response, is associated with a fatal outcome. These results provide new insight into early viral dynamics and the host response to LASV infection according to disease outcome.

## Author summary

Lassa fever, induced by Lassa virus, is a viral hemorrhagic fever of great concern as it is endemic to a large part of Africa, with tens of thousands of cases per year, and there is no effective treatment or licensed vaccine. Lassa virus can be handled only in biosafety level 4 facilities and access to patients is limited. As a result, little is known about the pathogenesis of this severe disease. Non-human primates, and macaques in particular, are the most relevant animal models for studies of this disease. We previously developed a cynomolgus monkey model reproducing the different outcomes of Lassa fever observed in humans:

**Data Availability Statement:** We have deposited our transcriptomic data at the following URL: https://zenodo.org/records/14163706. All other

relevant data are in the manuscript and its supporting information files.

**Funding:** This study was funded by a grant to SB from The Délégation Générale pour l'Armement (Agence Nationale de la Recherche - Accompagnement Spécifique des Travaux de Recherches et d'Innovation Défense, ANR-ASTRID 2014, France), a grant to SB from the Fondation pour la Recherche Médicale (FRM, France), and by a PhD grant attributed to JH from the Ecole Normale Supérieure de Lyon. The funders had no role in study design, data collection and analysis, decision to publish, or preparation of the manuscript.

**Competing interests:** The authors have declared that no competing interests exist.

limited clinical signs and recovery versus catastrophic illness and death. Here, we use this model to investigate further the dynamics of Lassa virus dissemination, cell tropism in different organs, and host response to infection. Our findings demonstrate that cell tropism and viral dissemination differ considerably between outcomes, underlying the different host responses to Lassa virus infection and differences in severity.

## Introduction

Lassa fever is a viral hemorrhagic fever caused by Lassa virus (LASV), an Old-World mammarenavirus of the *Arenaviridae* family [1]. The disease is endemic to West Africa, particularly Nigeria, Benin, Guinea, Sierra Leone, and Liberia. In these countries, and more particularly in Nigeria and Benin, outbreaks of Lassa fever occur frequently, with lethality rates ranging from 15 to 30% [2,3]. Sporadic cases are also detected in neighboring countries, such as Ghana, Togo, Côte d'Ivoire, Mali, and Burkina Faso [4–6]. LASV is transmitted to humans through contact with its natural reservoir, the peridomestic rodent *Mastomys natalensis*, or material contaminated with the excreta of these rodents, but alternative reservoirs have also been described [7]. No vaccine has been licensed to date, and the only antiviral drug available, ribavirin, has a very low efficacy or is completely ineffective [8,9]. Lassa fever is, therefore, a major public health issue in West Africa and has been listed by the WHO as an epidemic threat requiring urgent preparedness. The severity of Lassa fever ranges from asymptomatic infection in a large proportion of patients, to a fatal viral hemorrhagic fever. The disease begins with nonspecific signs, such as fever, asthenia, myalgia, headache, anorexia, abdominal pain, sore throat, diarrhea, vomiting and nausea. Facial and/or neck edema, conjunctivitis, hemorrhages, neurological signs, and acute respiratory distress are then observed in patients with severe disease, and death occurs in a context of multiorgan failure [10]. Patients surviving acute disease may experience sequelae, such as pericarditis, deafness and ataxia [11,12].

LASV is a viral species displaying genetic diversity, with seven lineages described to date, corresponding to different areas of circulation of the virus [13]. Lineages 1, 2, and 3 circulate in Nigeria, lineage 4 is found in Guinea, Sierra Leone, and Liberia, lineage 5 has been described in Mali and Côte d'Ivoire, and lineage 7 predominates in Togo and Benin [4,14–16]. By contrast, lineage 6 has been isolated only from rodents [7]. All LASV strains can cause fatal disease in humans, and it remains unclear whether there are differences in the pathogenicity of the various strains in patients. However, in the cynomolgus monkey model, the different lineages seem to induce disease with different severities [17–19]. As both patient access and the handling of LASV patient samples are very limited, animal models are crucial to understand the pathogenesis of Lassa fever and the events enabling some patients to control LASV and leading to catastrophic illness and death in others. LASV has a preferential tropism for antigen-presenting cells (APC), dendritic cells and macrophages, potentially accounting for the rapid transfer of the virus from site of local infection to secondary lymphoid organs (SLOs) [20,21]. However, this tropism does not seem to favor the induction of LASV-specific T-cell responses [22–24]. The replication of LASV in SLOs is followed by systemic dissemination to almost all organs during severe Lassa fever in cynomolgus monkeys, whereas the spread of the virus appears to be restricted to a few organs in surviving animals [17,20,25]. We and others have shown that LASV strains belonging to lineage 5, such as AV and Soromba [5,16], cause less severe disease than the Josiah strain in cynomolgus monkeys [18,26]. We have recently developed a cynomolgus monkey model that reproduces the different outcomes of Lassa fever observed in humans [25]. Infecting animals with the Josiah and AV strains of LASV at the

same dose and via the same route of inoculation leads to uniformly fatal Lassa fever and non-severe disease and survival, respectively. We have shown that, following its initial replication at the inoculation site, LASV reaches the SLOs, where it displays significant levels of replication in all animals. However, the dynamics of viral dissemination then differ strikingly between outcomes. LASV displays no significant further spread in surviving animals, whereas systemic viral dissemination and replication is observed in animals with fatal disease, leading to a cytokine/chemokine storm and multiple organ failure closely resembling that observed in septic shock syndrome. Here, we further analyzed the dynamics and tropism of these two strains of LASV in these animals, to shed light on the pathogenesis of Lassa fever and the immune mechanisms associated with the different outcomes.

## Results

### Dynamics of LASV tropism in cynomolgus monkeys

The clinical signs observed in these animals after AV and Josiah Lassa virus infection, as well as the survival curves, have been previously described and are not presented here [25]. Histopathological features of AV and Josiah infection have also been previously reported for these animals thanks to hematoxylin-eosin stained tissue sections [25]. Two days post infection (DPI), a few infected cells were detected by *in situ* hybridization (ISH) in the subcortical sinus of the inguinal lymph nodes (ILN) draining the injection site in both AV- (Fig 1A) and Josiah-(Fig 1B) infected cynomolgus monkeys. A massive infection of the ILN and MLN paracortex was detected 5 DPI in both groups of LASV-infected animals. At this time point, only a few infection foci were detected in the splenic red pulp of Josiah-infected animals, and to a lesser extent, in that of AV-infected animals (S1 Fig). By 11 DPI, Josiah and AV LASV had spread beyond the previously infected areas to reach the germinal centers of the LN and splenic white pulp. During in the recovery of AV-infected animals (28 DPI), residual viral RNA was detected in the germinal centers and white pulp. Multiplex immunostaining was performed to identify the LASV-infected cells. In the MLN of AV-infected animals, LASV GPC was detected mostly in CD68+ cells (monocytes or macrophages) at 5 and 11 DPI (Fig 1C). A massive infiltration of neutrophils (calprotectin+ cells) was observed at 5 DPI in the MLN of Josiah-infected animals, and LASV antigen colocalized mostly with CD68+ calprotectin+ cells and desmin+ cells, corresponding to stromal cells, and, to a lesser extent, CD68+ cells, at 5 and 11 DPI. In the spleens of AV- and Josiah-infected animals at 5 and 11 DPI, LASV GPC was detected in CD68+, calprotectin+, and desmin+ cells, and neutrophil infiltration was observed (Fig 1D). T lymphocytes (CD3+ cells) were often found close to infected cells in the spleens of AV-infected animals. In the livers of AV-infected animals, viral RNA was detected only in immune cell infiltrates and in sinuses and not until 11 DPI (S1 Fig). In Josiah-infected animals, viral RNA was detected as early as 5 DPI, and a massive infection with immune cell infiltrates, affecting Kupffer cells and hepatocytes, was observed at 11 DPI. AV LASV RNA was not detectable at any time point in the kidneys, lungs, cerebellum, and heart. Only a few AV LASV RNA-positive cells were detected in the brain, always within capillaries. By contrast, large amounts of Josiah LASV RNA were detected in kidney endothelial cells at 11 DPI. In the lungs of Josiah-infected animals, viral RNA was present in mononuclear cells from 5 DPI onwards, extending to epithelial cells by 11 DPI. In Josiah-infected animals, a massive infection of the endothelial cells of the brain and cerebellum was observed (S2 Fig). A few infected cells were detected in the heart in Josiah-infected animals. The adrenal glands were a preferential site of Josiah LASV replication, with numerous clusters of infection observed in the medulla, and more particularly in the cortex, on day 11. By contrast, only immune cells were infected in the adrenal glands of AV-infected animals, and no viral replication was observed in parenchymal cells. Massive LASV

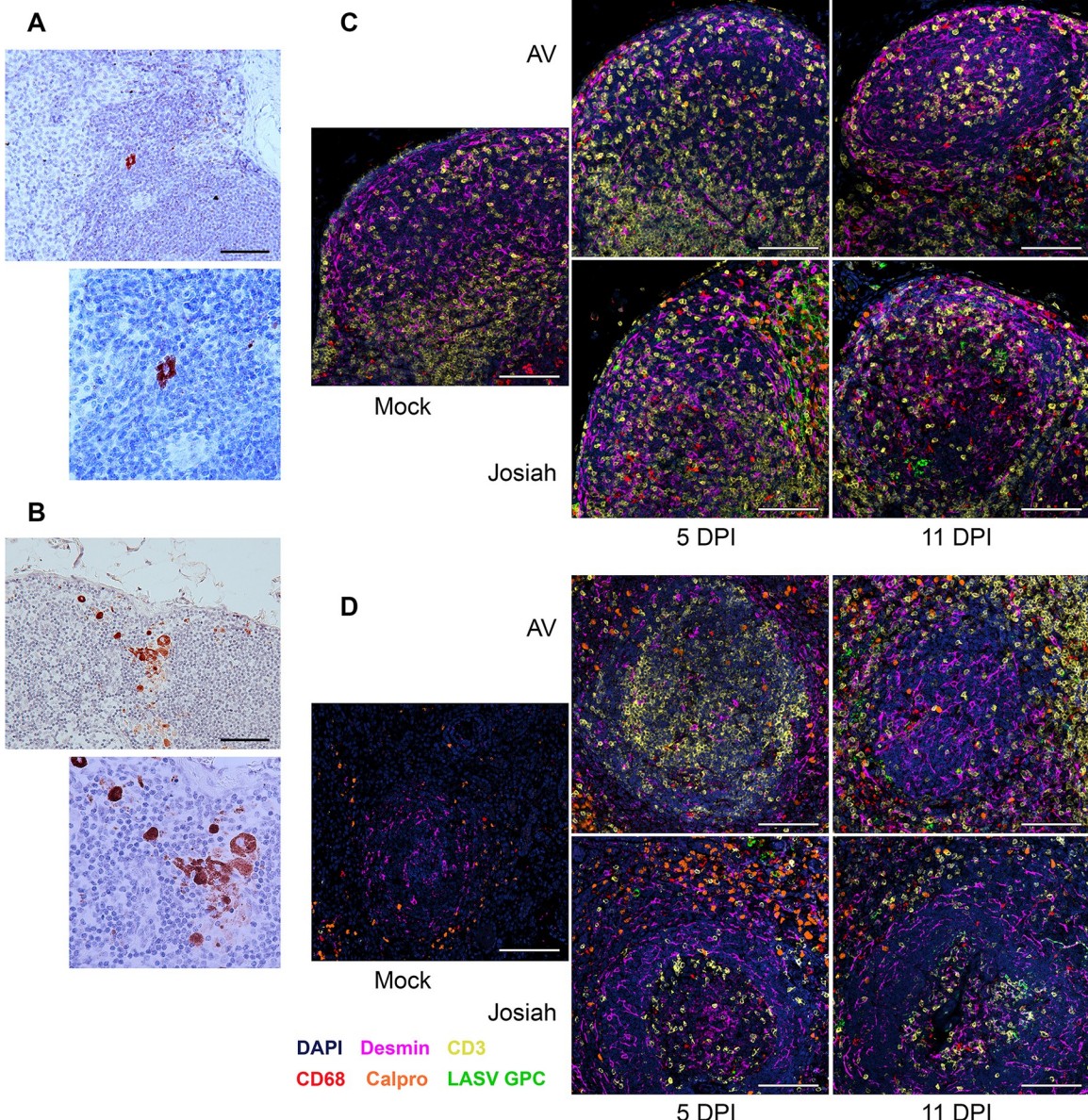

**Fig 1. LASV tropism in SLOs.** Detection of LASV RNA (red) by ISH in the ILN of animals infected with AV (A) or Josiah (B), with hematoxylin staining. Scale bars: 100 μm. Identification of LASV target cells by immunofluorescence in the MLN (C) and spleens (D) of AV- and Josiah-infected animals on days 5 (left image) and 11 (right image) after infection, and mock animals, by confocal microscopy. The following stains were used: LASV GPC (green), calprotectin (orange), CD68 (red), CD3 (yellow), desmin (pink), and DAPI (blue). Scale bars: 100 μm.

replication was observed in the thymus in all infected animals at 11DPI, and viral RNA was already present on day 5 in Josiah-infected monkeys (data not available for AV-infected animals). In this tissue, endothelial cells and epithelial cells, but also cells not identified by the panel used, were infected by Josiah LASV (S3A Fig). AV LASV mRNA continued to be detected during recovery in surviving animals. Numerous infected cells were detected in the pancreas 11 DPI in Josiah-infected animals (data not available for AV-infected animals). Intensive LASV replication occurs in the intestine. At 5 DPI, infected cells were detected within gut-associated lymphoid tissue (GALT) in both AV- and Josiah-infected animals, but

were found in close contact with intestinal villi only in AV-infected animals. At 11 DPI, numerous intestinal epithelial cells were infected in both groups. Finally, no viral RNA was detected in the ovaries at 5 DPI, in either AV- or Josiah-infected animals, or at 11 DPI in AV-infected females (no Josiah-infected females were analyzed at 11 DPI). At 11 DPI, Josiah LASV RNA was found in the endothelial cells in the testis (no data available for the AV strain), whereas during recovery (28 DPI), clusters of infected cells were found in the testis, including the seminiferous tubules.

## Identification of LASV targets and inflammation in non-lymphoid tissues

We characterized the phenotypes of infected cells in non-lymphoid tissues obtained at 11 DPI from both AV- and Josiah-infected animals (Fig 2). Numerous hepatocytes and some macrophages (CD68$^+$) replicating Josiah LASV were identified in liver tissue, whereas AV LASV GPC was detected only within CD68$^+$ cells located in hepatic sinusoids, corresponding to Kupffer cells. We observed an increase in the density of T cells and macrophages in the liver in AV-infected animals. In the presence of Josiah LASV, these infiltrations were more intense, particularly for T cells, and were accompanied by numerous neutrophils. Other immune cells negative for the phenotypes tested were also detected in these animals. In animals with fatal infections, viral RNA colocalized with CD31 in the kidneys, indicating an infection of endothelial cells. Massive morphological alterations were observed in the lungs of Josiah-infected animals. Considerable thickening of the alveolar wall was observed, together with moderate infiltration by macrophages and a large number of T cells. LASV GPC was detected in neutrophils, macrophages, endothelial cells, and epithelial cells, this last group of cells being identified on the basis of their location. By contrast, no infected cells were detected in the lungs of AV-infected animals at the peak of the disease, and no significant inflammation was noted. Epithelial cells, and a few endothelial cells, were targeted by Josiah LASV in pancreas (S3B Fig). During Josiah LASV infection, the infected cells in the brain were mostly endothelial cells, whereas no AV LASV RNA was found in the brain in surviving animals. An activation of microglia (IBA1$^+$) and a clustering of these cells around infected endothelial cells, was characteristic of Josiah virus infection in this organ. T cells seemed to accumulate within blood vessels surrounded by activated microglial cells. We further analyzed LASV tropism in the brains of Josiah-infected animals. We found that the viral genome was localized mostly to vessel walls in the parenchyma and along the meninges and subarachnoid space. One area of intense viral staining was the choroid plexus, in which we identified infected endothelial cells lining the blood vessels and infected ependymocytes—large cuboidal epithelial cells that secrete cerebrospinal fluid (Fig 3A). The infected cells in the brain parenchyma were CD31$^+$ endothelial cells on the luminal side of the vessel wall and Iba1$^+$ pericytes on the abluminal side of the vessel (Fig 3B). We found no evidence of active replication in neuronal layers or of colocalization of the viral genome with the GFAP marker expressed by glial cells. We quantified the neutrophils (calprotectin$^+$) and of cytotoxic cells (GrzB$^+$) present in the different organs (Fig 4). We observed a significant increase in the number of GrzB$^+$ cells in lungs from Josiah-infected animals at 11 DPI, but these cells were not CD3$^+$ (Fig 4A, 4B and 4C). No significant change in the number of GrzB$^+$ cells was observed in liver, kidneys, brain, adrenal glands, or thymus in infected animals during the disease (Fig 4C). A significant neutrophil infiltration was observed in the lungs, kidneys, and adrenal glands 11 days after infection with Josiah LASV but not after infection with AV LASV (Fig 4A, 4B and 4C). In the brains of Josiah-infected animals, such an infiltration was already present at 5 DPI but had increased by day 11, whereas only a moderate but significant increase in neutrophil number was detected 11 days after AV LASV infection (Fig 4C).

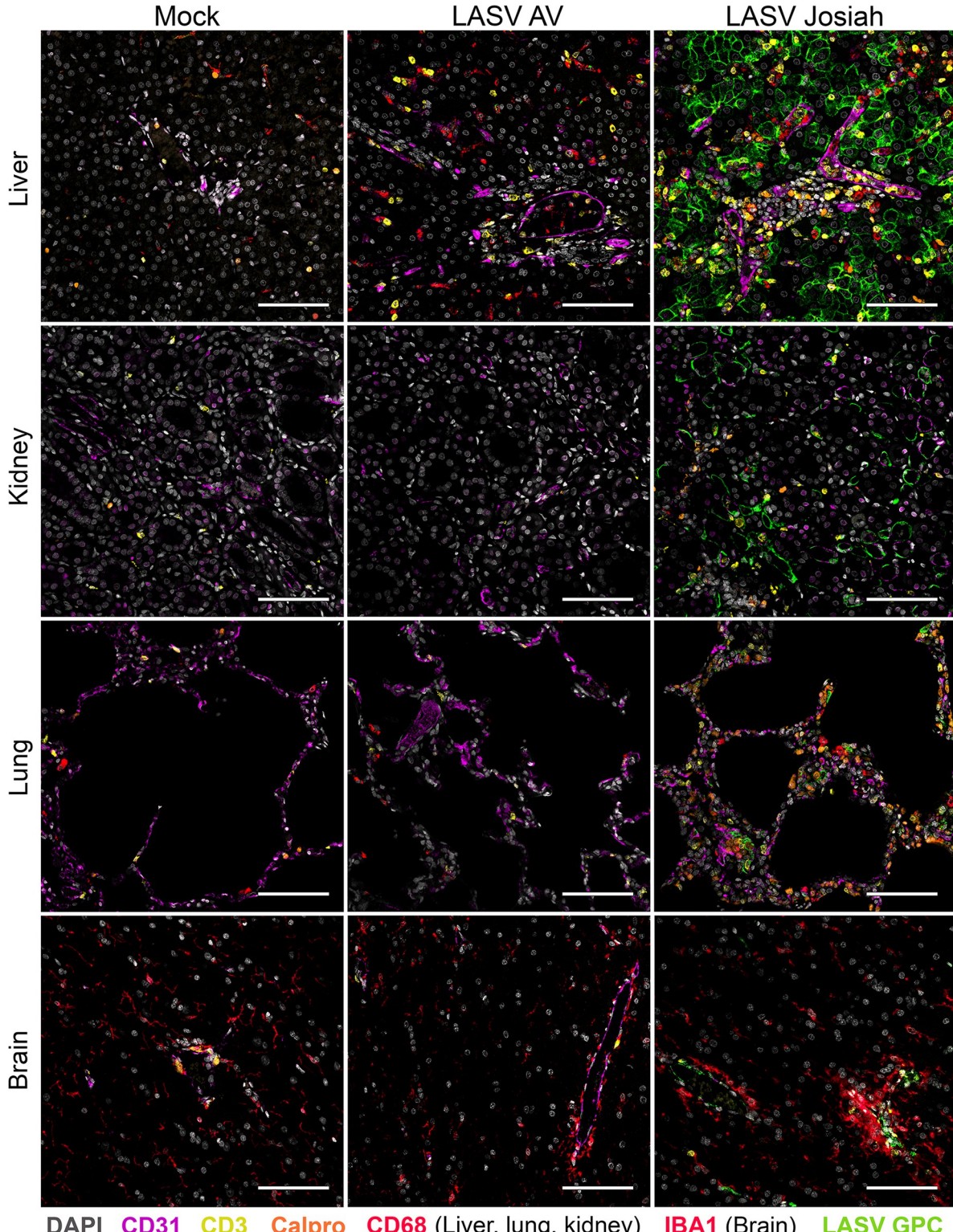

**Fig 2. LASV tropism in organs.** Liver, kidney, lung, and brain tissues obtained at 11 DPI from mock-, AV-, and Josiah-infected animals were stained for LASV GPC (green), calprotectin (orange), CD68 or IBA1 (red), CD3 (yellow), desmin (magenta), and with DAPI (blue) and analyzed by confocal microscopy. Scale bars: 100 μm.

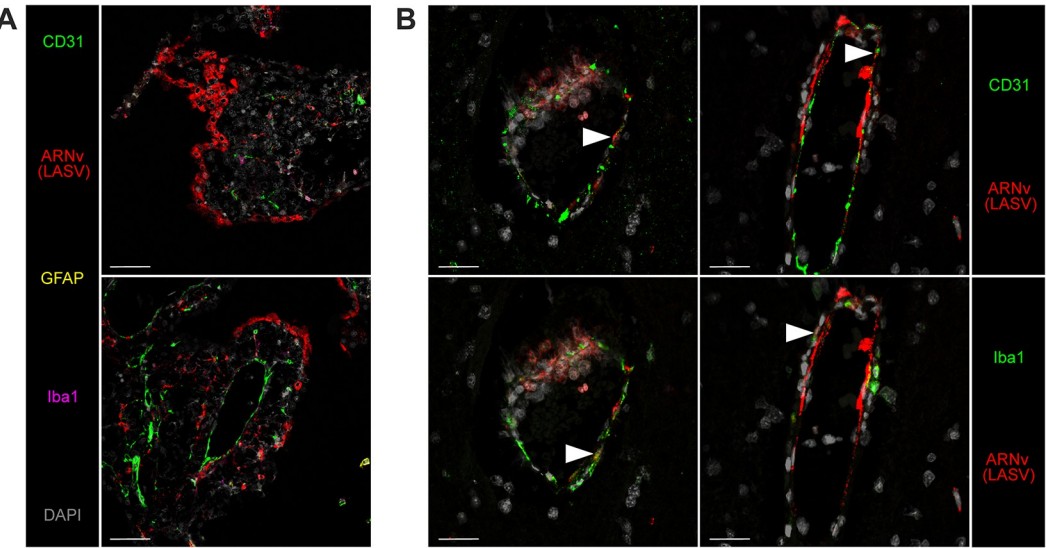

**Fig 3. Josiah LASV tropism in the brain.** Brain sections were obtained from Josiah-infected animals at 11 DPI and analyzed by confocal microscopy. A. The choroid plexus was stained for LASV RNA (red), CD31 (green), GFAP (yellow), Iba1 (magenta), and with DAPI (gray). The bottom image shows the histological structure of the plexus, which contains, from the center out: the blood vessel lumen, endothelium, connective tissue, ependymal cells and the ventricle lumen. Scale = 50 μm. B. Intraparenchymal capillaries stained for LASV RNA (red) and CD31 (green, upper images) or Iba1 (green, lower images). The arrows indicate LASV-infected endothelial cells (upper images) and pericytes (lower images). Scale = 20 μm.

## Analysis of cell content in the organs of infected animals based on transcriptome deconvolution

We investigated the change in cell content in several compartments during LASV infection, using the CIBERSORT/CIBERSORTx method for deconvolution [27]. A significant decrease in the proportion of memory B cells was observed in the LN and spleen 10 days after LASV infection, followed by an increase at day 28 in AV-infected animals (Fig 5A). An enrichment in plasma cells was noted in the spleens of Josiah-infected animals at the terminal stage of the disease. AV-infected animals presented a high proportion of CD8+ T cells in the spleen on day 10, with only non-significant changes observed in other samples and in the LN. An enrichment in resting memory CD4+ T cells was observed on days 2 and 28 in the LN of AV-infected animals, whereas a decrease in the proportion of these cells was observed in the spleen in infected animals. A strong enrichment in activated memory CD4+ T cells was observed on day 10 in the LN of infected animals whereas the proportion of follicular auxiliary and regulatory T cells (Treg) decreased concomitantly in these organs. An increase in the number of activated NK cells—significant in the LN and non-significant in the spleen—was detected 10 DPI in AV-infected animals. The proportions of monocytes and M1 macrophages were found to have increased at day 5 in AV- and Josiah-infected animals, respectively, whereas an enrichment in M1 macrophages was not detected until 10 DPI in the spleens of AV-infected animals. Finally, the proportion of activated mastocytes had increased by day 5 in the LN of AV-infected animals, whereas eosinophil levels in these organs decreased from day 10 onwards in both groups.

The same approach was applied to PBMCs (Fig 5B). The proportion of naive B cells had increased and that of memory B cells had decreased at day 10 in Josiah-infected animals. A non-significant increase in the proportion of plasma cells during the course of the disease was observed in all infected animals. A decrease in the proportion of CD8+ T cells was observed in both groups. An enrichment in resting memory CD4+ T cells was observed at day 4 in AV-

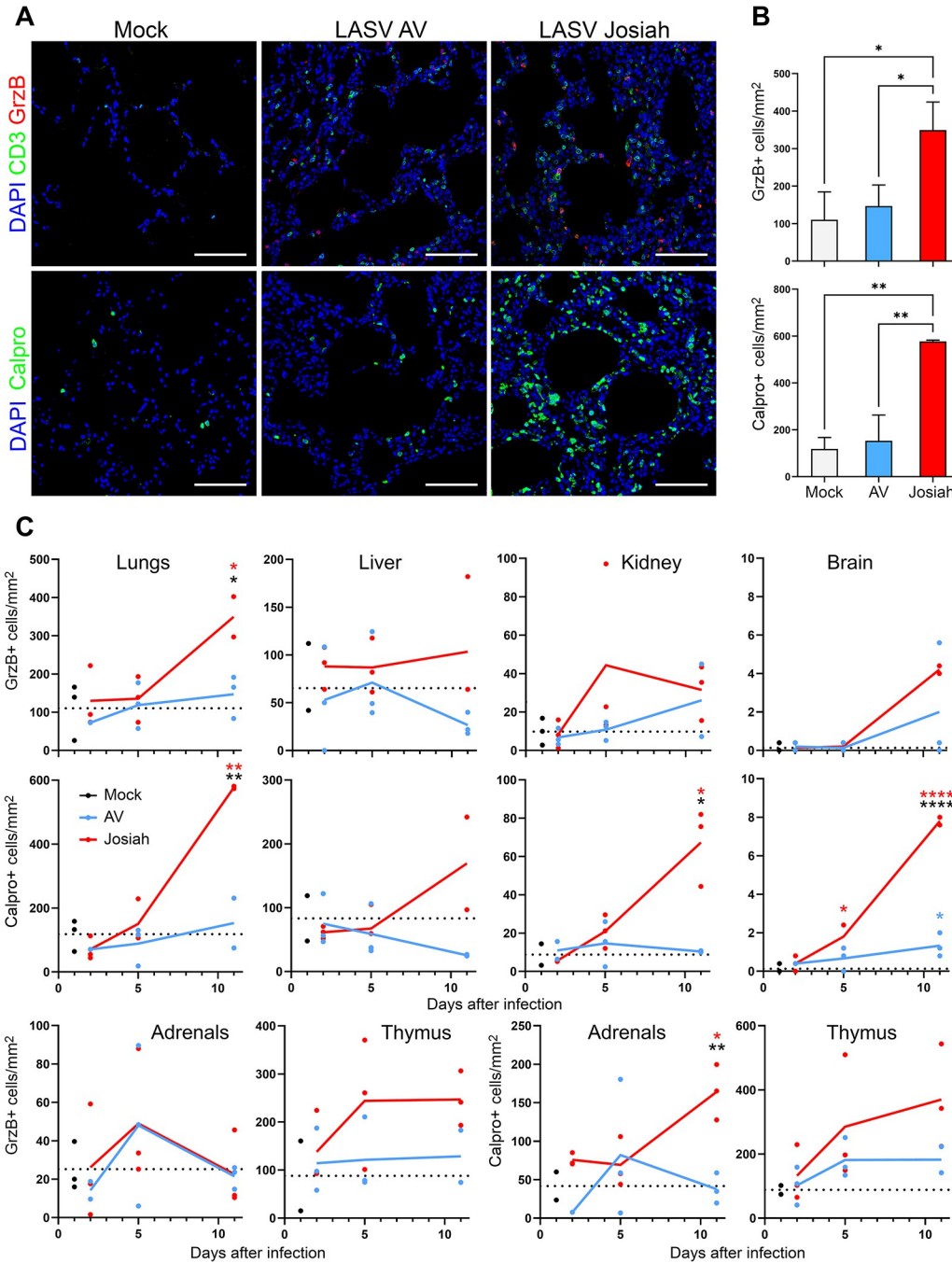

**Fig 4. Quantification of the cytotoxic cells and neutrophils infiltrating the lungs.** (A) Staining of lungs obtained from mock-infected animals and AV- or Josiah-infected animals ($n = 3$) at 11 DPI for CD3 (green), GrzB (red), and with DAPI (blue) (upper images) or for calprotectin (green) and with DAPI (blue) (lower images). Scale bars: 100 μm. (B) Quantification of GrzB+ cells (upper graph) or calprotectin+ cells (lower graph) in the lungs at 11 DPI. Results are expressed as the mean of three individual values ± standard error of the mean (SEM) of the number of cells/mm$^2$, with * indicating $p < 0.05$ and ** $p < 0.01$. (C) Quantification of GrzB+ cells (upper graph) or calprotectin+ cells (lower graph) in the lungs, liver, kidneys, brain, adrenal glands, and thymus at 2, 5, and 11 DPI in AV- (blue) and Josiah- (red) infected animals ($n = 3$ per group). Results are expressed as the mean (line) and individual values, in numbers of cells/mm$^2$, for LASV-infected animals. The mean value for the mock-infected animals ($n = 3$) is indicated by the dotted line and individual values are shown in black. The colored asterisks indicate significant differences between Josiah- or AV-infected animals and mock-infected animals, black asterisks indicate significant differences between Josiah- and AV-infected animals. * indicates $p < 0.05$, ** $p < 0.01$ and *** $p < 0.001$.

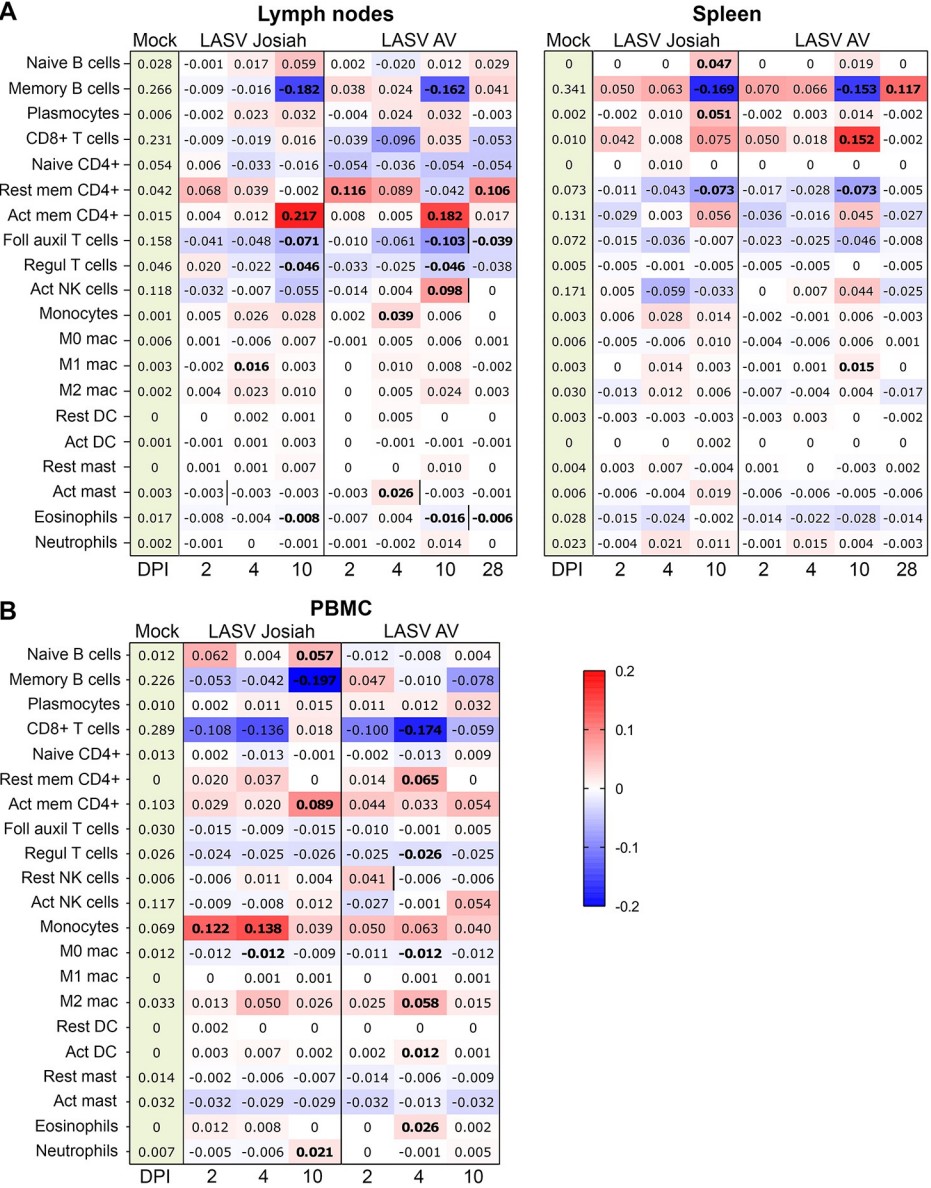

**Fig 5. Cell content determination by transcriptome deconvolution in SLOs and PBMCs.** RNA-seq data obtained with MLN and spleen (A) or with PBMCs (B) from mock-infected animals or LASV-infected animals at 2, 5, 10, and 28 (AV only) DPI were used for a deconvolution analysis by the CIBERSORT method with the LM22 matrix signature ($n = 3$ for each group). The values in the "mock" column indicate the mean proportion of each cell type within the total cell population. The values indicated in the other columns are the differences of the mean value for the group concerned relative to that for mock-infected animals. These differences are also illustrated with the colorscale in a heatmap. Significant differences ($p < 0.05$) with respect to the mock-infected animals are indicated by numbers in bold typeface, and differences between Josiah-infected and AV-infected animals at a given time point are indicated by a vertical black line.

infected PBMCs, and an enrichment in activated memory CD4[+] T cells was observed at day 10 in Josiah-infected animals. The proportion of Treg decreased during the course of the disease in both groups. An enrichment in monocytes was observed in both groups and was significant in the group infected with Josiah, whereas the proportion of M0 macrophages dropped transiently at day 4 in all animals, this decrease being accompanied by an increase in the proportion of M2 macrophages. A significant enrichment in activated DCs and eosinophils was

observed four days after AV infection, and a significant enrichment in neutrophils was observed at day 10 in Josiah-infected animals. However, given that transcriptomic profiles were determined for cells obtained after the crushing of the LN and spleen, the proportions of adherent cells, macrophages, DC and, polynuclear cells obtained may not be reliable.

## Transcriptomic profile of cells derived from the splenic marginal zone at the peak of Lassa fever

The splenic marginal zone (SMZ) is a crucial area for immune responses. We therefore investigated the transcriptomic profile of SMZ-derived cells purified from FFPE sections by laser-capture microdissection 11 DPI. The viral load in this area was significantly higher in Josiah-infected animals than in AV-infected animals (Fig 6A). We performed a gene set enrichment analysis with the HALLMARK database and identified pathways for which there were significant differences between Josiah- and AV-infected animals (Fig 6B). The IFNα and γ response, E2F target, IL6/JAK/STAT3, G2/M checkpoint, IL2/STAT5 and complement pathways were activated during Lassa fever, as shown by comparison with mock-infected animals, but this activation was more intense after Josiah LASV infection. The MYC targets V1, unfolded protein response, and oxidative phosphorylation pathways were activated in Josiah-infected animals relative to AV-infected animals, but were downregulated in AV-infected animals relative to mock-infected animals. The mTORC1 signaling pathway was upregulated in Josiah-infected animals relative to AV-infected animals. Finally, the apoptosis pathway was upregulated in Josiah-infected animals relative to all other groups. We then analyzed the expression of gene sets related to different immune parameters and observed that those specific to monocytes, the type I IFN response, B cells, and cytokines were activated during LASV infection, and that this activation was stronger for Josiah LASV (Fig 6C and 6D). The T cell-related gene set was also activated after LASV infection, but with no difference between LASV strains.

## Transcriptomic responses in hepatocytes at the peak of Lassa fever

We used the same approach to determine the transcriptome of hepatocytes purified 11 DPI by laser-capture microdissection from FFPE sections stained for LASV GP. We purified non-infected and LASV-infected hepatocytes from Josiah-infected animals, whereas only non-infected cells were collected after AV infection, as no LASV antigen was detected in the livers of AV-infected animals (Fig 7). The number of viral reads in each cell population was consistent with status (Fig 7A). An analysis of pathway activation (HALLMARK) showed an activation of the IFNα and γ responses after LASV infection, which was more intense with Josiah LASV than with AV LASV and in infected hepatocytes than in non-infected hepatocytes (Fig 7B). A similar pattern was observed for allograft rejection, the IL6/JAK/STAT3 pathway, the inflammatory response, and complement pathways, but with less marked differences. The mitotic spindle, UV response, mTORC1 signaling, protein secretion, epithelial mesenchymal transition, MYC targets V1, G2M checkpoint, PI3K AKT mTOR signaling, E2F targets, apoptosis, TGFβ signaling, heme metabolism, apical junction, glycolysis, and unfolded protein response pathways were downregulated in LASV-infected animals relative to control animals, and this downregulation was more marked for AV than for Josiah. However, in Josiah-infected animals, these pathways were upregulated in infected hepatocytes relative to uninfected hepatocytes. We then analyzed the expression of certain gene sets in these cells. The type I IFN response was strongly upregulated in Josiah-infected hepatocytes relative to non-infected hepatocytes and hepatocytes purified from mock- and AV-infected animals (Fig 7C and 7D). The gene set related to cytokines, chemokines and their receptors was activated during LASV infection, whatever the viral strain and infection status of the hepatocytes. Only a few genes

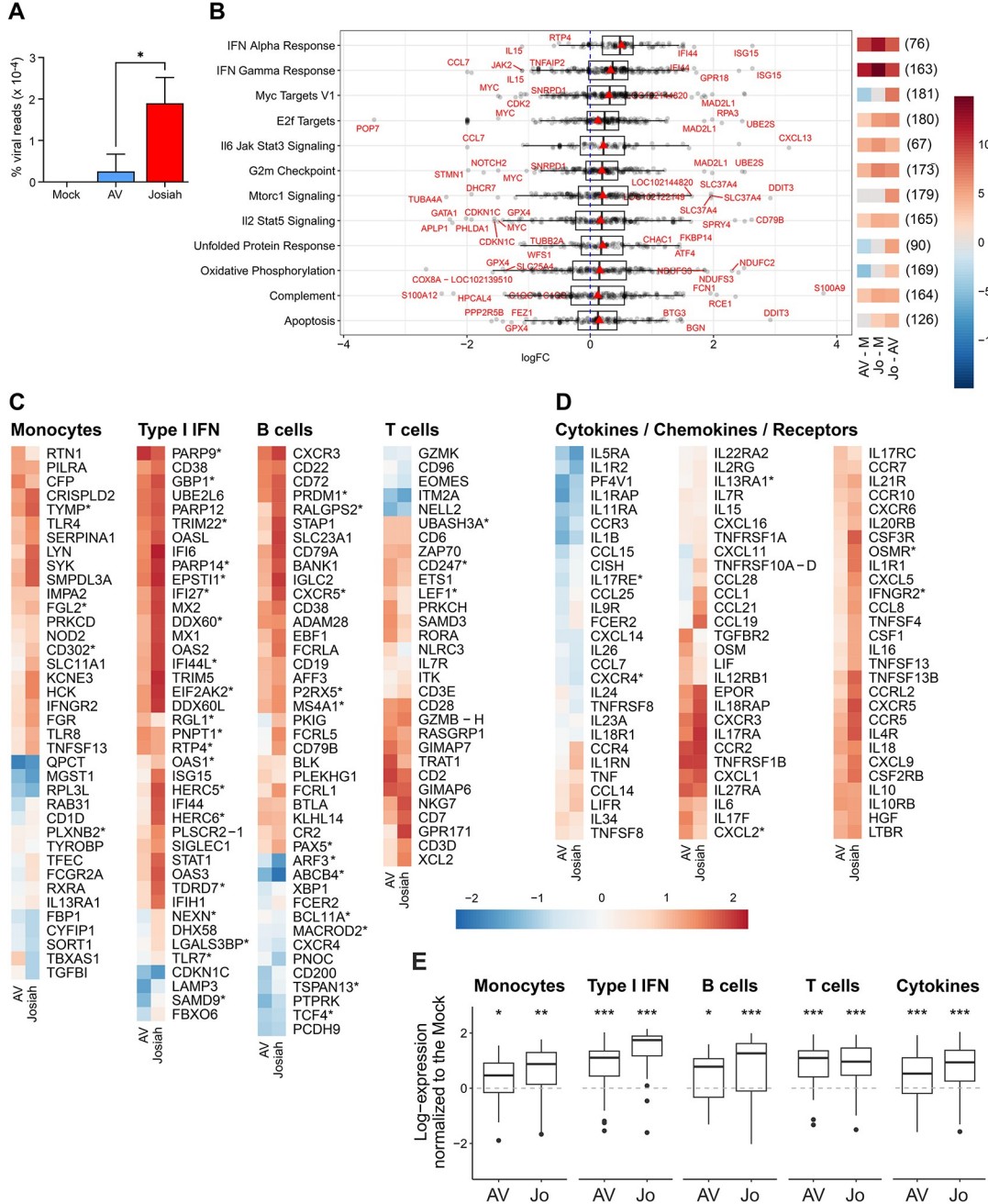

**Fig 6. Transcriptomic profile of splenic marginal zone cells.** (A) The percentage of total sequence reads mapping on LASV genome is represented as the mean ± SEM for splenic marginal zone cells purified by LCM at 11 DPI from mock-, AV-, and Josiah-infected animals ($n = 3$ per group). * indicates a significant difference ($p < 0.05$) between groups. (B) Differential expression of genes from certain HALLMARK pathways expressed as a log2 fold-change (logFC) for Josiah-infected splenic marginal zone cells obtained at 11 DPI compared with AV-infected cells. Each individual gene is represented by a dot and the total number of genes in the pathway is indicated in parentheses. The genes distribution is summarized by boxplots in black and the average differential expression by red triangles. Genes with extreme logFC (2*standard deviation of the logFC) are labeled with their names in red. Heatmaps of the CAMERA enrichment scores for the pathways are presented for each comparison between animal groups: AV-infected animals vs. mock-infected, Josiah-infected vs. mock-infected, and Josiah-infected vs. AV-infected. Red indicates a global upregulation of the pathway, blue a global downregulation, and the color intensity the amplitude of the variation. (C) Heatmaps representing gene expression for the monocyte, type I IFN, B-cell and T-cell gene sets and (D) for the gene set for cytokines/chemokines/receptors. The colors on the heatmap represent the mean standardized (centered and scaled) gene expression for AV- and Josiah-infected splenic marginal zone cells obtained at 11 DPI ($n = 3$ per group) normalized against that in mock-infected samples ($n = 3$). The intensity of expression is indicated by the scale. Gene names with

an * indicates significance for any of the comparisons (E) Boxplot representation of the distribution of the log expression of the genes in each pathway between AV- and Josiah-infected splenic marginal zone cells obtained at 11 DPI (*n* = 3 per group) normalized against the mock-infected samples (*n* = 3) (central line, median; limits, first and third quartiles; whiskers, largest or smallest value no more than 1.5 times the interquartile range away from the hinge). Outlying data are plotted individually.

displayed significant differential expression between the two coagulation-related gene sets, leading to only moderate alterations for the first gene set and an absence of modification for the second gene set. For the first of these gene sets, an upregulation was observed in Josiah-infected hepatocytes relative to non-infected hepatocytes, whereas a downregulation was observed in Josiah-infected hepatocytes relative to AV-infected animals. The transcriptomic response in the livers of these animals has already been reported elsewhere [25]. We performed a transcriptome deconvolution analysis to characterize the change in cell populations in the liver during LF (S4A Fig). Some of the changes observed were similar for all outcomes. These changes included a decrease in hepatocyte numbers and proportions, and an increase in the proportions of hepatocytes, plasma cells, non-inflammatory macrophages and T cells. However, an increase of the proportion of periportal and central vein CESFs and of inflammatory macrophages was observed 11 DPI in Josiah-infected animals.

## Transcriptomic profile in the lungs of cynomolgus monkeys during Lassa fever

We established the transcriptomic profiles of the lungs at the peak of the disease by RNA sequencing (RNA-seq). LASV infection induced dramatic changes in the patterns of transcription in the lungs, and the mRNA profiles of Josiah-infected animals were different from those of AV-infected animals 11 DPI (Figs 8A and S5A). Transcriptome deconvolution analysis suggested an enrichment in immune cells and, to a lesser extent, in endothelial cells, in the lungs of Josiah-infected animals relative to the animals of the other groups (Figs 8B and S4B). We observed an enrichment in effector memory CD4+ T cells, NK cells, myeloid dendritic cells (DCs), classical macrophages, and megacaryocytes, whereas AV-infected animals presented a significant enrichment only for macrophages and plasmacytes. An increase in the proportions of aerocytes and capillaries was observed in Josiah-infected lungs, together with a decrease in the proportions of vascular smooth muscle, bronchial vessel, and alveolar epithelial cells. AV infection seems to result in the overexpression of a larger number of genes than Josiah infection (Fig 8C). Type I IFN response-related genes were only weakly activated at peak disease after AV infection whereas these genes were strongly overexpressed in Josiah-infected animals (Figs 8D and S6A). Monocyte-related genes were upregulated in Josiah-infected animals relative to AV-infected animals, in which these genes were downregulated relative to mock-infected animals. T cell-related genes were activated in both groups of infected animals, but they were more strongly expressed, with a different pattern in Josiah-infected animals relative to AV-infected animals. Some genes were overexpressed in both groups of infected animals relative to mock-infected animals. These genes included *CD96*, *CD247*, *LCK*, *ITK*, *ZAP70*, *CD3D*, *CD6*, *SH2D2A*, *UBASH3A*, *CD28*, *CD7*, *TRAT1*, *GIMAP7*, and *NLRC3*. Expression levels for *ITM2A* and *NELL2* were lower in Josiah-infected animals than in the other groups, whereas a large number of genes were upregulated in Josiah-infected animals relative to the other groups: *EOMES*, *GIMAP6*, *PRF1*, *TIGIT*, *RASGRP1*, *GRP171*, *GZMA/B/K*, *CD2*, *CD3ε*, *SIRPG*, and *TRAC*. Finally, some genes were overexpressed in AV-infected animals relative to the other groups. These genes included *PRKCH*, *RORA*, *NKG7*, *GATA3*, *GZMM*, *LEF1*, *XCL1/2*, *SAMD3*, and *KLRB1*. Cytokine mRNAs were present at higher levels in infected animals, and were higher in Josiah-infected animals than in AV-infected animals. The levels of a few

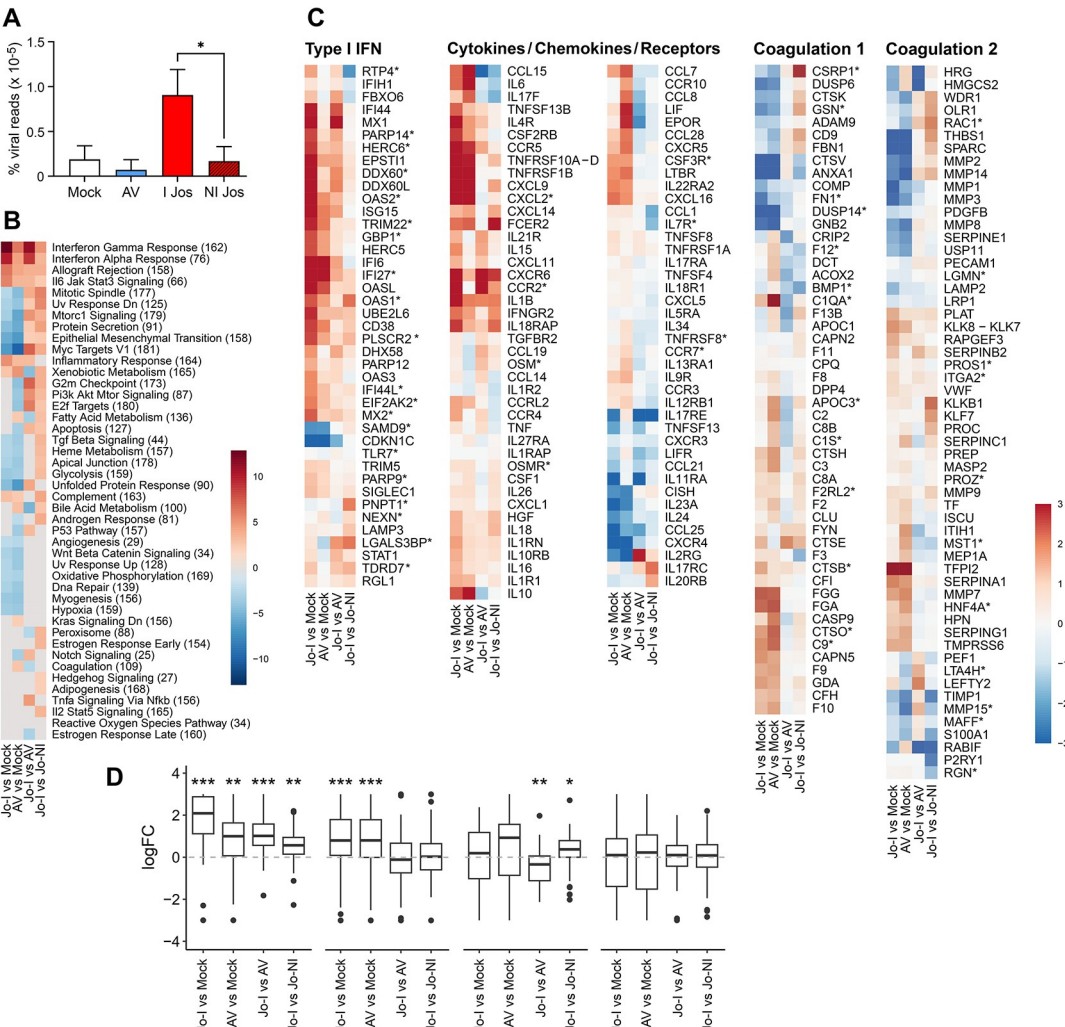

**Fig 7. Transcriptomic profile of hepatocytes.** (A) The percentage of total sequence reads mapping on LASV genome is represented as the mean ± SEM for hepatocytes purified by LCM at 11 DPI from mock-, AV-, and Josiah-infected animals (*n* = 3 per group). We purified non-infected (NI) and infected (I) hepatocytes from the Josiah-infected animals. * indicates a significant difference (*p* < 0.05) between conditions. (B) Heatmaps of CAMERA enrichment scores for the most significant pathways in gene set analyses applied to the HALLMARK gene sets. Comparisons between Josiah-infected hepatocytes (Jo-I) and mock-infected hepatocytes, AV-infected hepatocytes (AV) and mock-infected hepatocytes, Josiah-infected hepatocytes and AV-infected hepatocytes, and Josiah-infected hepatocytes and non-infected hepatocytes from Josiah-infected animals are shown on the *x*-axis with gene sets on the *y*-axis. The number of genes in each pathway is indicated in parentheses. Gray squares indicate gene sets displaying no significant regulation, whereas the red and blue squares indicate pathways significantly upregulated (red squares) or downregulated (blue squares) in one group relative to the other. (C) Heatmaps representing gene expression variations for the type I IFN, cytokines/chemokines/receptors, and coagulation gene sets. The colors on the heatmaps represents the differential expression (DE) of genes between hepatocytes obtained from animals in different groups at 11 DPI (*n* = 3 per group). The comparisons are the same as for (B). The DE of genes is expressed as a log fold-change value and is given by the color scale. Red (resp. blue) indicates an upregulation (resp. downregulation) of the gene in the group at the numerator of the comparison compared to the group at the denominator. (D) Boxplot representation of the log FC values showing the variation in the expression of the genes in each pathway between groups of hepatocytes obtained at 11 DPI (*n* = 3 per group) (central line, median; limits, first and third quartiles; whiskers, largest or smallest value no more than 1.5 times the interquartile range away from the hinge). Outlying data are plotted individually.

mRNAs, such as the *LIFR*, *EPOR*, *TNFRSF*, *IL-34*, and *IL-11RA* mRNAs, were lower in Josiah-infected animals than in the other groups, but most of the genes from this gene set were over-expressed in Josiah-infected animals. For chemokine-related mRNAs, we observed an increase in *CXCL9* and *CXCR3/6* mRNA levels in infected animals, and in *CCR3* and *CCRL2* mRNA

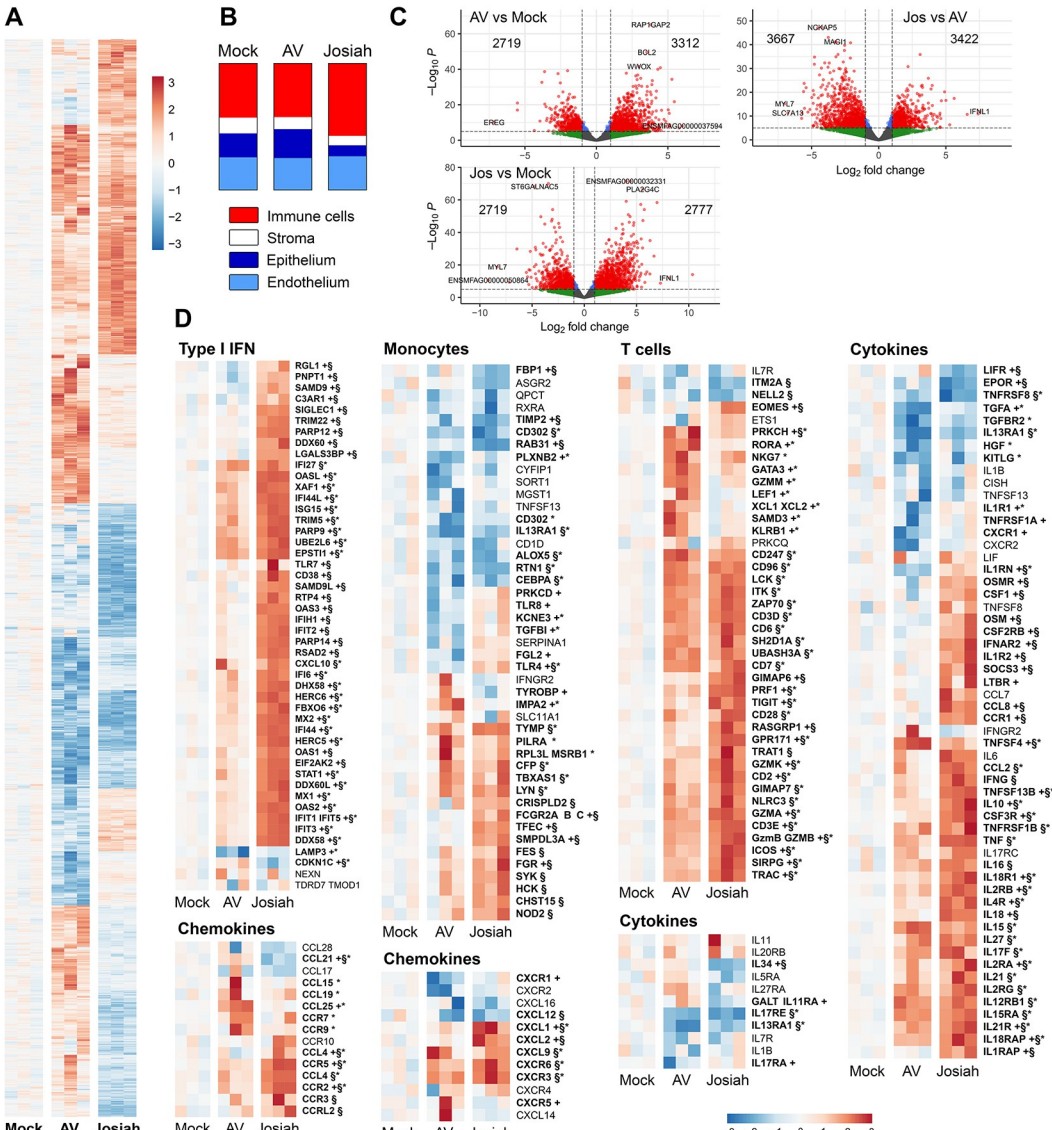

**Fig 8. Transcriptomic profile of the lungs.** (A) Significant changes in global gene expression (absolute log FC > 1 and adjusted p-value < 0.05) in the lungs of mock-, AV-, and Josiah-infected animals at 11 DPI. Individual values from three animals per group are presented. The color indicates the standardized (centered and scaled) gene expression, normalized against that in mock-infected samples. The intensity of expression is indicated by the color scale. Red (resp. blue) indicates expression higher (resp. lower) to the mock-infected mean. (B) The relative proportions of pulmonary cell types in the lungs at 11 DPI were determined by transcriptome deconvolution and grouped by functional type. The values presented were obtained from the data in S2B Fig. (C) Volcano plots representing the genes differentially expressed (red circles) (threshold of adjusted p-value < 0.05 and absolute log2FC > 1) between two groups, as indicated in the table. The x-axis reflects the log2 fold-change difference in expression for each gene in the first group with respect to the second group. The y-axis indicates the statistical significance of these results, calculated as the–(minus) Log of the p-values. The numbers of downregulated and upregulated differentially expressed genes for each comparison are shown in the upper left and right parts of the graphs, respectively. Blue dots represent differentially expressed genes with absolute log2FC < 1, green and gray dots represent not differentially expressed genes with absolute log2FC > 1 for green and < 1 for gray. (D) Heatmaps representing standardized gene expression for the type I IFN, monocytes T-cell, cytokine, and chemokine gene sets in lung extracts obtained at 11 DPI from AV- and Josiah-infected animals. Individual values from three animals per group are presented. The color indicates the standardized (centered and scaled) gene expression, normalized against that in mock-infected samples. The intensity of expression is indicated by the color scale. Red (resp. blue) indicates expression higher (resp. lower) to the mock-infected mean. Gene names with a symbol indicates significance for each of the following comparison * for AV-infected animals compared to mock-infected, + for Josiah-infected animals compared to AV-infected animals and § for Josiah-infected animals compared to mock-infected.

levels in the Josiah-infected animals only. By contrast, *CCL15/19/25* and *CCR5/7* mRNA levels were upregulated only in AV-infected animals. Finally, *CCL21* and *CXCL12* mRNA levels were downregulated and *CCL4*, *CXCL1/2/9*, and *CCR2/5* mRNA levels were upregulated in Josiah-infected animals relative to the other groups. We investigated whether the transcriptome profile of the lungs was consistent with that for other acute pulmonary syndromes, by analyzing the gene sets reported to be up- (S7A Fig left) or down- (S7A Fig, right) regulated in acute pulmonary lesions [28] and up- (S7B Fig) or down- (S7C Fig) regulated in SARS-CoV-2 infection [29]. We found that 20 of 42 genes from the acute pulmonary syndrome gene sets were upregulated and 14 of 42 were downregulated in Josiah-infected animals, whereas 34 of 160 and 78 of 139 genes from the SARS-CoV-2 gene sets were down- and upregulated, respectively in the same animals.

## Transcriptomic profile in the kidneys of cynomolgus monkeys during Lassa fever

We had access to RNA samples obtained from kidneys at 5 and 11 DPI for RNA-seq analysis. Transcriptomic profiles at 5 DPI were similar between infected animals, regardless of viral strain, but mRNA levels tended to return to basal levels by day 11 in AV-infected animals (Fig 9A and 9B). The principal component analysis plot was consistent with these observations (S5B Fig). By contrast, major changes in mRNA levels were observed at 11 DPI in Josiah-infected animals. No significant change in cell proportions was detected in the kidneys on transcriptome deconvolution analysis at 5 DPI in infected animals, with the exception of a modest decrease in neutrophil content in AV-infected animals (Figs 9C and S4C). At 11 DPI, the proportion of glomerular capillaries was lower in Josiah-infected animals than in AV-infected animals and the proportions of the principal cells were lower in Josiah-infected animals than in the other groups of animals. An increase in the proportion of pelvis epithelial cells and neutrophils was also observed in Josiah-infected animals. Massive type I IFN-related gene expression was observed in infected animals at 5 DPI, returning to basal levels at 11 DPI in AV-infected animals but remaining high at this time point in Josiah-infected animals (Figs 9D and S6B). For monocyte-derived genes, only samples obtained from Josiah-infected animals presented significant changes in gene expression, with a decrease in the expression of *TFEC*, *CD302*, *HNMT*, *SLC11A1*, and *TIMP2* and an increase in mRNA levels for *IFNGR2*, *TNFSF13*, *SERPINA1*, *PILRA*, *CFP*, *TYMP*, *NOD2*, *FCRGR2*, *LYN*, *FRS*, *FGR*, *TLR4*, *LY75*, *TEXAS1*, and *HCK*. For T-cell-related mRNAs, strong expression was observed in Josiah-infected animals at 11 DPI, when most of these genes were overexpressed, with the exception of *ITM2A*, which was downregulated. By contrast, T-cell mRNAs were overexpressed only at 5 DPI in AV-infected animals. The cytokine- and chemokine-related gene sets followed the same pattern, with most genes strongly expressed at 11 DPI with the exception of *EFOR* in Josiah-infected animals and *CXCL14* in AV-infected animals. In AV-infected animals, these mRNAs were synthesized earlier, being detected at 5 DPI, but at moderate levels.

## Transcriptomic profile in the adrenal glands of cynomolgus monkeys during Lassa fever

The adrenal glands were among the most massively infected organs in our animals, and infected cells were found in all areas of these glands, with the zona fasciculata most strongly affected (Fig 10A). We investigated the possible effect of LASV tropism for the adrenal glands on their endocrine properties, by quantifying corticoids involved in homeostasis (cortisol) and arterial tension (aldosterone), and renin in plasma (Fig 10B). An increase in aldosterone and cortisol concentrations was observed 12 DPI with the Josiah strain, and these animals also had

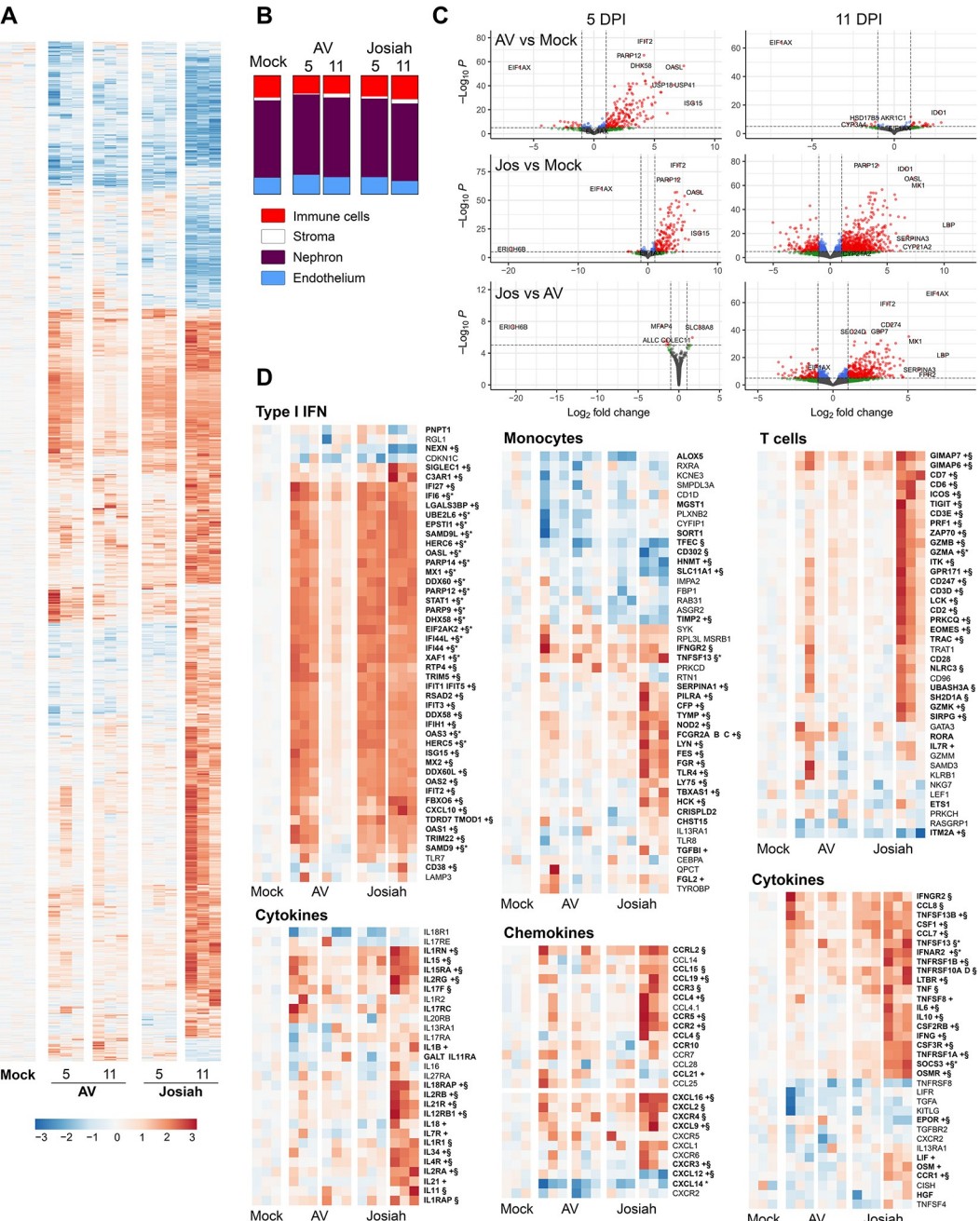

**Fig 9. Transcriptomic profile of the kidneys.** (A) Significant changes in global gene expression (absolute log FC > 1 and adjusted p-value < 0.05) in the kidneys of mock-infected animals and those of AV-, and Josiah-infected animals at 5 and 11 DPI. Expressed as in Fig 8A. (B) Relative proportions of cell types in the kidneys of mock animals and LASV-infected animals at 5 and 11 DPI, determined by transcriptome deconvolution and grouped by functional type. The values presented were obtained from the data in S2A Fig. (C) Volcano plots representing the genes differentially expressed as for Fig 8C between AV-infected and mock-infected animals (upper plots), Josiah-infected and mock-infected animals (center plots), and Josiah-infected and Av-infected animals (lower plots). Comparisons are presented for 5 (left plots) and 11 DPI (right plots). (D) Heatmaps representing standardized gene expression for the type I IFN, monocyte, T-cell, cytokine, and chemokine gene sets, as for Fig 8D, except that gene expression was assessed at 5 (left columns) and 11 (right columns) DPI for LASV-infected animals.

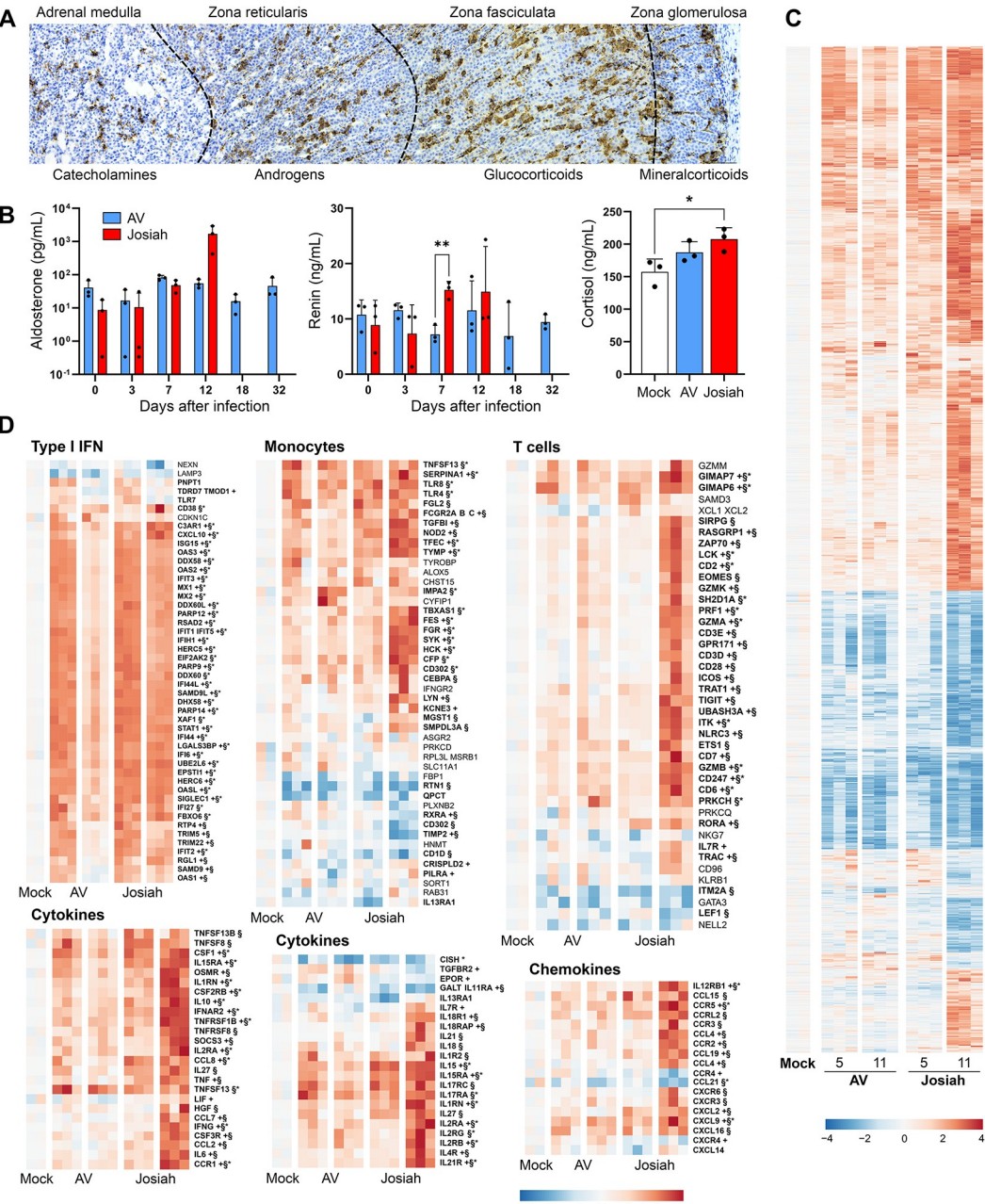

**Fig 10. Adrenal glands and LASV infection.** (A) Tropism of LASV for the adrenal glands, by functional zone. ISH techniques were used to stain LASV RNA in the adrenal glands of Josiah-infected animals at 11 DPI. The various zones of the adrenal glands are indicated above the image and the families of hormones secreted by each zone are indicated below. (B) Quantification of adrenal hormones in the plasma of animals. Levels of aldosterone in pg/mL, and of renin in ng/mL, in AV- and Josiah-infected animals, by time after infection. Levels of cortisol in ng/mL in mock-infected animals and in AV- and Josiah-infected animals at 12 DPI. * indicates $p < 0.05$, ** $p < 0.01$. (C) Significant changes in global gene expression (absolute log FC > 1 and adjusted p-value < 0.05) in the adrenal glands of mock-infected animals and in those of AV-infected and Josiah-infected animals at 5 and 11 DPI. Expressed as in Fig 8A. (D) Heatmaps representing standardized gene expression for the type I IFN, monocyte, T-cell, cytokine, and chemokine gene sets, as in Fig 9D.

higher levels of renin at 7 DPI. We determined the transcriptomic profiles of the adrenal glands at 5 and 11 DPI. However, only two samples were available for mock-infected animals. Similar changes in the expression of differentially expressed genes were observed in all infected animals at 5 DPI (Figs 10C and S5C). However, the pattern of expression remained unchanged at 11 DPI in AV-infected animals whereas the changes in expression were stronger at 11 DPI in Josiah-infected animals. Genes related to type I IFN responses strongly expressed as early as 5 DPI in infected animals, but their expression levels returned to normal by 11 DPI in AV-infected animals (Figs 10D and S6C). A similar pattern was observed for cytokine- and chemokine-related genes and for monocyte-related genes, except that the expression of *RTN1*, *RXRA*, *CD302*, *TIMP2*, and *CD1D* was downregulated at 11 DPI in Josiah-infected animals. Changes in the expression of T cell-related genes were observed principally at 11 DPI in Josiah-infected animals, which displayed an overexpression of this gene set.

## Discussion

The aim of this study was to characterize in more detail the dynamics of the pathogenesis and immune responses associated with survival or death from LF in a cynomolgus monkey model in which infection with AV LASV leads to non-lethal disease whereas infection with Josiah LASV results in death [25]. We previously showed that early LASV replication occurs in the skin adjacent to the subcutaneous site of infection. We show here that the draining lymph nodes are the first organ targeted by LASV after this early local replication, with infected cells detected in the subcortical sinus of ILNs close to the inoculation site at 2 DPI. This observation suggests that LASV probably accesses the LN draining the inoculation site through afferent lymph vessels, via infected APCs or free viral particles. We were unable to detect LASV in other tissues/organs at this time. The virus subsequently replicates in the ILNs, MLNs, GALT, and thymus and then in the spleen. These results confirm that lymphoid organs are an early major reservoir for LASV replication whatever the outcome of the disease, and that APCs—or at least macrophages—probably play a key role in the dissemination of LASV from the inoculation site. However, differences in tropism were observed between LASV strains in MLN, with only macrophages infected by the AV strain whereas stromal cells—probably fibroblastic reticular cells (FRC)—and neutrophils were also found to contain Josiah LASV material. From 5 DPI, tropism differed considerably between AV and Josiah LASV. Josiah LASV disseminated to almost all the organs, as previously reported for fatal LF in cynomolgus monkeys [17,20,26,30], replicating in non-immune cells, stromal cells, endothelial cells, hepatocytes, and epithelial cells. By contrast, AV LASV spread was limited to a few organs—the liver, adrenal glands and capillaries in the brain—with its effects limited essentially to immune cell infiltrates. This is surprising, as AV LASV is present in blood during the course of the disease, albeit in smaller quantities than Josiah LASV, and could therefore spread to tissues [25]. Endothelial cells are massively infected during infection with Josiah but not during infection with AV infection. The preferential tropism of LASV for endothelial cells is well known [17,20,25,31,32] and is probably crucial for the pantropism of LASV, enabling it to reach almost all organs. The reason for the lack of significant endothelial cell infection by AV LASV despite substantial viremia is unclear, but the lack of infection of these cells may account for the limited dissemination to peripheral tissues/organs. It would be interesting to determine using in vitro models whether endothelial cells present a different susceptibility and response to AV and Josiah infection. However, both strains are probably able to infected endothelial cells, as we previously showed that Mopeia virus, an Old-World arenavirus closely related to LASV, and AV LASV infect human umbilical vein endothelial cells (HUVEC) with similar replication kinetics and high titers of virus released in the supernatant [32]. A difference in

endothelial cell permeability between AV and Josiah-infected animals may also facilitate viral dissemination. This is particularly true for the brain in Josiah-infected animals, in which the infection of endothelial cells may enable the virus to enter the brain and to replicate in specific cells, such as ependymocytes and pericytes, particularly in the meninges, subarachnoid space, and choroid plexus. Some of these findings have been reported before [20]. This viral replication in the brain leads to local inflammation and meningoencephalitis and can explain the neurological signs observed in severely ill animals. The high levels of TNFα, IL-2 and IL-6 in the bloodstream of Josiah-infected animals [25], and of TNFα and IL-6 mRNA in their organs, at peak disease may contribute to vascular permeability, as previously described [33–36]. However, as endothelial cell infection and cytokine release occur late in infection, these events may exacerbate viral dissemination in organs rather than initiating it. Fatal LF was also associated with a massive infiltration of immune cells—neutrophils, macrophages, T cells, and probably NK cells—in most tissues, including those of the brain. Macrophage infiltration may also participate in the systemic dissemination of Josiah LASV. Numerous neutrophils infiltrated the lungs, adrenal glands, brains, and kidneys of Josiah-, but not AV-infected animals. Transcriptome deconvolution analysis also demonstrated neutrophil infiltration in the kidney. The presence of chemokine mRNA in the tissues of Josiah-infected animals is consistent with the observed infiltrations by macrophages, with high levels of CCL2 and CCL7 mRNAs, and neutrophils, with CCL2, CCL4, CXCL1 and CXCL2 mRNAs [37–39]. Moreover, CCL2 and CCL4 levels in the bloodstream were high at peak disease in Josiah-infected animals [25]. Transcripts for some chemokines involved in macrophage attraction, such as CCL19, CCL21, CCL25, and CXCL10, were detected in some tissues from AV- and Josiah-infected animals.

Neutrophils may play a role in pathogenesis during fatal LF, as these cells are known to cause tissue damage in the context of an exacerbated inflammatory response [40,41]. The transcriptome deconvolution analysis suggested an infiltration of plasma cells and macrophages into the lungs of AV-infected animals, whereas Josiah-infected animals had large proportions of CD4$^+$ T cells, NK cells, mDCs, and monocytes in the lungs. Significant infiltration by cytotoxic cells, probably NK cells, was also observed in Josiah-infected lungs. Very high levels of mRNAs related to the type I IFN response, monocyte and T-cell activation, and to cytokines and chemokines was observed in this organ during the terminal stage of Josiah LASV infection. These transcriptional changes are reminiscent of those observed in acute pulmonary lesions and in severe SARS-CoV2 infection. Together with the dramatic thickening of the alveolar walls [25], these changes may explain the acute respiratory distress syndrome observed in fatal LF. In AV-infected animals, the transcriptomic profile of the lung revealed only a moderate activation of T cells and the transcription of a few cytokine genes. Encephalitis is frequently associated with severe LF and neurological signs are observed in most Josiah-infected cynomolgus monkeys. The pathological changes observed in the brains of Josiah-infected animals are consistent with encephalitis, with infiltrations of inflammatory and cytotoxic cells, microglial activation and appearance of microglial nodules. Microglial nodules have been described during numerous acute and chronic infections and are associated with encephalitis [42,43]. We previously reported similar changes with other LASV strains causing severe disease in this model [17]. Our results suggest that Josiah LASV infiltrates the brain via the infection of capillary endothelial cells, before replicating in pericytes. These events were not observed with LASV AV. Liver failure is also a hallmark of fatal LF, and hepatocytes are a major target for LASV, with this tropism leading to a robust local inflammatory response [25]. We have previously shown that alterations to liver function during severe LASV infection are involved in the coagulopathy [44]. We show here that the synthesis of mRNAs related to the type I IFN response, cytokines and chemokines, and to coagulation pathways is induced in both infected and bystander hepatocytes in Josiah-infected animals and that this expression is also observed

in hepatocytes from AV-infected animals even without the infection of these cells. This observation suggests that liver inflammation results from the host inflammatory response rather than being induced directly by the viral infection of hepatocytes. Renal failure is associated with a poor prognosis for LF in both humans and in cynomolgus monkeys [2,25,45]. Massive expression of type I response-related genes was observed in the kidneys in both groups of animals at 5 DPI, demonstrating that this activation was not due to LASV tropism for this organ as no infection was detected in this organ after AV infection. This transcriptomic profile persisted 11 DPI in Josiah-infected kidneys, as did the expression of genes related to monocyte and T-cell activation and to cytokines and chemokines. At this stage, the massive viral replication and infiltrations with immune cells—mostly neutrophils—are probably the cause of this inflammatory response in the kidneys. The deconvolution analyses suggest that the proportions of the principal cells decreased significantly during acute Josiah LASV infection whereas the opposite pattern was observed with AV LASV. As these cells stimulate the reabsorption of $Na^+$ ions in response to aldosterone, their disappearance may be involved in the severe hyponatremia observed in Josiah-infected animals [25]. Our results confirm that the adrenal glands are a site of intense Josiah LASV replication [25]. The transcriptomic response in the adrenal glands was similar to that in the kidneys, suggesting that the same inflammation and infiltration occurred in these organs. All areas of the adrenal glands are infected, but the zona fasciculata is the most affected. LASV replication may alter the endocrine functions of the adrenal glands, with effects on the renin-angiotensin-aldosterone system, which plays a key role in controlling blood pressure, fluid balance and ionic composition. The increase in renin concentration in the blood of Josiah-, but not AV-infected animals is consistent with the pathogenicity of Josiah infection, as renin levels have been proposed as a marker of severity during shock [46]. This increase is probably induced by the low vascular pressure and hyponatremia characteristic of severe LF [25,30]. The high aldosterone levels observed at peak disease in fatal LF were probably induced in response to renin stimulation and therefore represent the physiological response induced to restore blood pressure. These observations suggest that the hypotension is not a consequence of alterations to liver/adrenal endocrine functions. The moderate but significant increase in cortisol levels in Josiah-infected animals probably reflects a stress response to infection and suggests that, despite the massive viral replication occurring in the adrenal glands, there are probably no major changes in the endocrine functions of these organs. The detection of LASV RNA in the testes of surviving animals one month after infection is consistent with the recent demonstration of the long-term persistence of the virus in testes and the potential risk for sexual transmission up to one year after acute disease [47].

Lymphoid organs are early major targets of LASV in these animals [25]. Our findings identify the draining LN as the first organ in which LASV replicates after initial replication at the inoculation site. By 5 DPI, the LN served as a reservoir for LASV dissemination, particularly with the Josiah strain. The lower viral loads in the LN, spleen, and thymus measured after infection with AV than after infection with Josiah [25] cannot explain the lack of dissemination of AV LASV to non-lymphoid organs, which is probably due to the induction of innate and/or adaptive immunity. The tropism of the LASV strains in LN differed, with AV antigens found only in APCs, whereas other cells such as stromal cells/FRC were also positive for Josiah LASV antigens. The reason for this discrepancy is unclear but it may have serious consequences for the course of the disease. Changes in the composition of the cell population, as determined by transcriptomic deconvolution, were similar for the two strains, for all but a few cell types. The SMZ is an important area for the presentation of blood-driven antigens, and macrophages present in this area have been shown to be crucial for the early control of lymphocytic choriomeningitis virus (LCMV) infection, another old-world arenavirus [48]. The higher viral load and more intense activation of gene sets related to immune activation in this

area 11 days after Josiah infection than after AV infection suggest that the SMZ is severely affected during fatal LF. This finding is reminiscent of the observations made in mice infected with LCMV, in which preferential viral tropism for SMZ macrophages leads to the induction of cytotoxic T lymphocytes (CTL) and the further destruction of this area by these CTL [49,50]. We observed an intense expression of T cell-related genes in the organs of Josiah-infected animals, but not in those of AV-infected animals, at 11 DPI. However, we never detected circulating LASV-specific T cells during the course of the disease in Josiah-infected cynomolgus monkeys [45,51,52]. These observations suggest that T-cell activation may be non-specific and that the widespread presence of activated bystander T cells may contribute to the disease, as previously suggested in patients with fatal infections [24]. One limitation of this study is the subcutaneous (SC) route of infection used, which is not similar to natural infection with LASV that occurs via respiratory tract. We cannot perform aerosol infection in our BSL4 facilities, and have choosen SC instead of intramuscular route to avoid a systemic route of infection. However, we think that this discrepancy with natural infection does not alter the course of infection, as lungs rapidly became targeted with the virus and similar pathological changes and viral tropism was reported in LASV-infected cynomolgus monkeys infected using aerosol route [30]. Another limitation is the fact that AV LASV has been isolated from a single patient who succumbed, showing that this virus can be fatal in humans whereas it is not in our model. However, as our aim is to model fatal and nonfatal Lassa fever in cynomolgus monkey, this discrepancy is not a problem.

This study shows that SLOs are early and crucial reservoirs for LASV replication in all infected animals, irrespective of the severity of the disease. Then, cell tropism, viral dissemination, and immune cell infiltration strikingly differ between animals according to the disease outcome. Moreover, our results suggest that events that take place in SLOs early during the course of infection may determine the later different host responses and the outcome of Lassa fever, either control of LASV and recovery or catastrophic illness and death. However, further experiments will be needed to characterize these mechanisms.

## Methods

### Ethics statement

All biological material was sampled from Josiah LASV-infected or AV LASV-infected cynomolgus monkeys during previous animal experiments [25]. All procedures have been approved by the Rhône Alpes Ethical Committee for Animal Experimentation (file number 2015062410456662, CECAPP, UMS3444/US8, Lyon, France).

### Tissue processing

Tissues and organs were harvested during necropsy on the animals and fixed by incubation in 4% formaldehyde solution for two weeks. Samples were dehydrated by passage through a series of increasing concentrations of ethanol (70–96%) and xylene and were then embedded in paraffin in a STP120 station (Microm-Microtech). Tissues and organs were then resized and embedded in paraffin on a TES99 machine (Tech-Inter) and cut with a microtome to obtain 3 or 10 µm-thick slices (for histological staining and laser microdissection, respectively), which were mounted on Super-Frost Plus (Fisher Scientific) microscope slides. Before further processing, slides were baked at 60°C for 1 h and dewaxed by incubation in Bond Dewax solution (Leica Biosystems) or in xylene.

### Histological staining

For chromogenic ISH, we performed heat-induced epitope retrieval (HIER) at 120°C with Target Retrieval solution (Biotechne) and ae Retriever 2100 pressure cooker (Aptum). Staining

was performed with the RNAscope 2.5 HD Detection kit Brown (Biotechne) according to the manufacturer's instructions and with probes targeting either the AV LASV GPC-coding sequence or the full-length Josiah LASV S segment (Biotechne). Sections on slides were then dehydrated in ethanol and xylene, mounted in Eukitt's xylene mounting medium (Sigma Aldrich) and covered with a coverslip. For multiplex staining (ISH and immunofluorescence), slides were stained with a Bond RxM stainer (Leica Microsystems) and the Opal 6-plex detection kit (Akoya Biosciences). Slides were subjected to up to six cycles of staining as follows: HIER was performed at 98˚C in Epitope Retrieval solution 1 or 2, at pH 6 or 9, respectively, for 30 minutes and the slides were then incubated with the primary antibodies against the following proteins: desmin (RB-9014-P1, Thermo Fisher Scientific), CD3 (A045229-2, Dako), CD68 (14–068882, Invitrogen), calprotectin (MA512213, Invitrogen), LASV GPC (mAb kindly provided by TG Ksiazek, PE Rollin, and P Jahrling (Special Pathogens Branch, Center for Disease Control, Atlanta, GA), CD31 (ab134168, Abcam), Iba1 (ab153696, Abcam), Granzyme B (ab237847, Abcam), and GFAP (ab68428, Abcam). A horseradish peroxidase (HRP-)coupled anti-mouse or anti-rabbit secondary antibody was then added, and binding was detected with an Opal fluorophore. The complex of primary and secondary antibodies was then stripped from the slide by HIER as previously described and another cycle of staining was performed. In multiplex panels including ISH, we performed the hybridization, amplification and detection of RNA before immunofluorescence staining, with the RNAscope 2.5 LS Multiplex Fluorescence Assay (Biotechne). Finally, sections were counterstained with spectral DAPI (Akoya Biosciences) mounted in Anti-Fade Fluorescence Mounting Medium (Abcam) and covered with a coverslip. Chromogenic staining was visualized under a DMIL microscope (Leica Microsystems) with LASX software, whereas fluorescence staining was assessed with an LSM980 confocal microscope (Zeiss) and ZEN software. For multiplex staining, the signals from different fluorophores were deconvoluted with the linear unmixing algorithm of ZEN. Images were analyzed with QuPath software.

### Laser-capture microdissection

For laser-capture microdissection, HIER was performed at 120˚C in the pressure cooker with a citrate solution at pH 6 (Sigma Aldrich). The sections were incubated with 3% bovine serum albumin solution for 30 minutes to block non-specific sites. The primary antibodies (CD68 for the MZ study and LASV GPC for hepatocytes) were then diluted in blocking solution and incubated with the sections at 4˚C overnight. Antibody binding was detected with an HRP-coupled secondary antibody (N-Histofine, Nichirei Biosciences) and amino-ethyl-carbazole and the sections were dehydrated in ethanol and xylene. Dissection was performed on a Pix-Cell II station (Arcturus Engineering) with Arcturus Capsure capsules (Macro LCM Caps, Applied Biosystems). Regions of interest were defined by visual assessment and laser pulses were used to create adhesion between the thermolabile capsules and the tissue to capture the cells. The DNA was eliminated by digestion and total RNA was extracted from the samples with the RNeasy Micro kit (Qiagen). For quantifications of the immune cells in infiltrates, we selected five individual fields of view at random for manual counting of the positive cells.

### RNA sequencing

Liver, lung, kidney and adrenal gland samples were treated with RNAlater (Thermo Fisher Scientific) and frozen for storage at -80˚C. After thawing, they were ground with a TissueLyser II (Qiagen) and clarified by centrifugation. Lymph nodes and spleens were crushed and passed through a filter with 70 μm pores. The red blood cells were lysed in ACK buffer (Thermo Fisher Scientific) and the isolated cells were frozen and stored at -150˚C. In both cases, total RNA was extracted using the RNeasy Mini kit after DNAse I treatment (Qiagen).

## Bioinformatic analysis of transcriptomic data

Bioinformatics analysis was performed using the RNAflow an internal pipeline (https://gitlab.pasteur.fr/hub/rnaflow). Reads were cleaned of adapter sequences and low-quality sequences using cutadapt. Only sequences at least 20 nt in length were considered for further analysis. STAR, with default parameters, was used for alignment on the reference genome. Genome was downloaded from UCSC and annotation track (.gtf Ensembl Genes) was retrieved from UCSC (genome: Crab-eating macaque, assembly Macaca_fascicularis_6.0). Genes were counted using featureCounts from Subreads package (parameters: -t gene -g ID -O -s 2). The statistical analysis was performed using R software. Differential analysis was performed using DESeq2 [DESEQ2] package. For organs study, genes with low reads counts were filtered using the filterByExpr function of edgeR package with default parameters. For LCM study, only null counts were filtered. A global DESeq2 model adjusted on the interaction of experimental condition and timepoints, and animal identifier when relevant was fit. Comparisons among groups were performed using the lfcShrink function with "ashr" option [ashr] and p-values were adjusted using the Benjamini-Hochberg multiple testing correction. The GSEA analysis was performed using the CAMERA method of the limma package [CAMERA]. The model was adjusted similarly to differential analysis model, with inter-gene correlation parameter set to 0. P-values were corrected using the Benjamini-Hochberg method. For single genes heatmaps, standardized gene expression is computed by normalizing the VST-transformed reads counts to the control group and scaled to a standard deviation of 1. Genes are clustered using the Ward.D2 aggregation criterion and the Euclidean distance. The GSEA heatmaps show the enrichment Z-score computed by CAMERA for each comparison between 2 experimental groups. These heatmaps integrate the significance level of each gene set as grey color indicates not significant genes sets with adjusted p-values greater to 5%. Genes sets are ranked according to their averaged Z-score across all comparisons.

## Deconvolution analysis of transcriptomic data

The evolution of the cell-type composition estimates based on the transcriptomic data of the organs was calculated from the bulk transcriptomic data using the CIBERSORT deconvolution method [CIBERSORT], as implemented in github.com/favilaco/deconv_benchmark. The method was applied to signatures specific to each tissue under study. The LM22 cell-type signature was used for SLO. For lungs, liver and kidneys, we used published single cell atlas and aggregated the single cell data by cell type to obtain matrix signatures [53–59]. We refer in the text to Baderlab for liver signature [liver], Krasnowlab for lungs signature [lungs], also available online (https://hlca.ds.czbiohub.org/) and kidneycellatlas for kidney signature [kidney].

## Quantification of corticosteroids

Plasma was isolated from blood samples collected into tubes containing EDTA and was frozen for storage at -80˚C. Thawed samples were processed in accordance with the manufacturer's instructions for the following kits: for aldosterone (KGE016, R&D Systems), cortisol (KGE008B R&D Systems), and renin (13530-AAT, Euromedex).

## Statistics

For counts of cells on histological sections and transcriptome deconvolution analyses, the different groups of animals were compared by ANOVA test with Tukey tests for multiple comparisons.

## Supporting information

**S1 Fig. Detection of LASV RNA in ILN, MLN, spleen, liver, kidney, lung and brain.** LASV RNA was detected in the various organs by ISH with AV- and Josiah-specific probes (brown). Hematoxylin staining is shown in blue. Samples obtained at 5 and 11 DPI from AV- infected and Josiah-infected animals were analyzed, together with samples obtained at 28 DPI for AV-infected animals. Scale bars: 100 μm.
(TIF)

**S2 Fig. Detection of LASV RNA in cerebellum, adrenals, thymus, pancreas, intestine, heart and gonads.** LASV RNA was detected in the various organs by ISH with AV- and Josiah-specific probes (brown). Hematoxylin staining is shown in blue. Samples obtained at 5 and 11 DPI from AV- infected and Josiah-infected animals were analyzed, together with samples obtained at 28 DPI for AV-infected animals. Scale bars: 100 μm.
(TIF)

**S3 Fig. LASV tropism in thymus and pancreas.** Thymus (A) and pancreas (B) sections obtained at 11 DPI from Josiah-infected animals were stained for LASV GPC (green), calprotectin (orange), CD68 (red), CD3 (yellow, not shown for the merge image of thymus), desmin (magenta), and with DAPI (blue) and analyzed by confocal microscopy. Scale bars: 100 μm.
(TIF)

**S4 Fig. Proportion of cell types in organs, as determined by transcriptome deconvolution.** RNA-seq data obtained for the liver (A), lungs (B), and kidneys (C) of mock-infected animals or LASV-infected animals at various time points after infection were used for a deconvolution analysis by the CIBERSORT method with a matrix signature appropriate to each tissue: Baderlab for liver [liver], Krasnowlab for lungs [lungs] and kidneycell atlas for kidneys [kidneys] ($n$ = 3 for each group). The values in the "mock" column indicate the mean proportion of each cell type within the total cell population. The values indicated in the other columns indicate the differences between the mean value for the group concerned and that for the mock-infected animals. These differences are also illustrated with the colorscale in a heatmap. Significant differences ($p < 0.05$) with respect to mock-infected animals are indicated by numbers in bold type, and differences between Josiah-infected and AV-infected animals at the same time point are indicated by a vertical black line.
(TIF)

**S5 Fig. Principal component analysis of the transcriptomic data in lungs, kidneys, and adrenal glands following infection with LASV deciphering sources of variations.** Principal component analyses were performed on the transcriptomic data for lungs (A), kidneys (B), and adrenal glands (C) on the 500 most variant genes. The figure shows the coordinates of samples for the two first principal components, for which the corresponding inertia is indicated in the parentheses in the $x$- and $y$-axis labels. Colors are attributed according to animal group and number of days after infection (B and C only).
(TIF)

**S6 Fig. Modulation of specific gene sets in the lungs, kidneys, and adrenal glands of LASV-infected animals.** Boxplot representation of the log FC values showing the variation in expression of the genes of the type I IFN, monocyte, T-cell, and cytokines/chemokines gene sets in the lungs (A), kidneys (B), and adrenal glands (C) between groups of hepatocytes obtained at 5 and 11 DPI ($n$ = 3 per group) (central line, median; limits, first and third quartiles; whiskers, largest or smallest value no more than 1.5 times the interquartile range away from the hinge). Outlying data are plotted individually. Comparisons are presented between Josiah-infected

animals (J), AV-infected animals (A), and mock-infected animals (M). For (A), only samples obtained at 11 DPI are presented. For (B) and (C), samples obtained at 5 and 11 DPI are presented.
(TIF)

**S7 Fig. Comparison of LASV-induced transcriptomic variations with the changes induced by other diseases.** (A) Heatmaps representing standardized gene expression of the genes reported to be upregulated (left columns) or downregulated (right columns) during acute pulmonary lesions. Heatmaps representing standardized gene expression of the genes reported to be upregulated (B) or downregulated (C) during SARS-CoV-2 infection. The colors on the heatmaps represents the standardized (centered and scaled) gene expression in lung extracts obtained at 11 DPI from AV- and Josiah-infected animals normalized against the gene expression in lung extracts from mock-infected animals. Individual values are represented for each group ($n$ = 3 per group). The intensity of gene regulation is indicated by the score scale. Red (resp. blue) indicates expression higher (resp. lower) to the mock-infected mean. Gene names with a symbol indicates significance for each of the following comparison * for AV-infected animals compared to mock-infected, + for Josiah-infected animals compared to AV-infected animals and § for Josiah-infected animals compared to mock-infected.
(TIF)

**S1 Data. xlsx file contains numerical data for Figs 4B, 4C and 10B.**
(XLSX)

# Acknowledgments

We thank L Barrot, S Barron, A Vallve, A Duthey, B Labrosse, D Pannetier, S Mély, D Thomas, S Godard, E Moissonnier, A Pocquet, J Valois, and C Léculier (P4 INSERM–Jean Mérieux, US003, INSERM) for assistance in conducting the BSL-4 experiments. We also thank G Jouvion for training and advice on immunohistochemical processing and staining. We thank S Becker (Institut of Virology, Marburg, Germany) for providing us with the LASV strains, and TG Ksiazek, PE Rollin, and P Jahrling (Special Pathogens Branch, Centers for Disease Control, Atlanta, GA) for the anti-LASV monoclonal antibodies.

# Author Contributions

**Conceptualization:** Sylvain Baize.

**Formal analysis:** Jimmy Hortion, Emeline Perthame, Sylvain Baize.

**Funding acquisition:** Sylvain Baize.

**Investigation:** Jimmy Hortion, Emeline Perthame, Blaise Lafoux, Laura Soyer, Stéphanie Reynard, Alexandra Journeaux, Clara Germain, Hélène Lopez-Maestre, Natalia Pietrosemoli, Nicolas Baillet, Séverine Croze, Catherine Rey, Catherine Legras-Lachuer, Sylvain Baize.

**Methodology:** Jimmy Hortion, Emeline Perthame, Blaise Lafoux, Stéphanie Reynard, Hélène Lopez-Maestre, Natalia Pietrosemoli, Catherine Legras-Lachuer, Sylvain Baize.

**Supervision:** Sylvain Baize.

**Validation:** Emeline Perthame, Natalia Pietrosemoli, Sylvain Baize.

**Visualization:** Jimmy Hortion, Emeline Perthame, Blaise Lafoux, Sylvain Baize.

**Writing – original draft:** Sylvain Baize.

**Writing – review & editing:** Emeline Perthame, Blaise Lafoux, Natalia Pietrosemoli, Sylvain Baize.

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
