## [Decision Letter · Decision Letter 0]

17 Oct 2024

Dear Dr. Baize,

Thank you very much for submitting your manuscript "Fatal Lassa fever in cynomolgus monkeys is associated with systemic viral dissemination and inflammation" for consideration at PLOS Pathogens. As with all papers reviewed by the journal, your manuscript was reviewed by members of the editorial board and by several independent reviewers. The reviewers appreciated the attention to an important topic. Based on the reviews, we are likely to accept this manuscript for publication, providing that you modify the manuscript according to the review recommendations.

This report provides a comprehensive comparative analysis of the tissue tropism, histopathological manifestations, and transcriptional changes following infection of cynomolgus macaques with two distinct strains of Lassa fever virus. Both reviewers were impressed by the scope of the study and the quality of the findings and are enthusiastic about the report. The reviewers provided comments that should help to clarify and improve the manuscript, which will likely primarily encompass edits and additions to the text.

We apologize for the length of time that it has taken to provide this feedback, which was due to one reviewer failing to return their comments.

Sincerely,

Allan J Zajac

Academic Editor

PLOS Pathogens

Thomas Hoenen

Section Editor

PLOS Pathogens

Michael Malim

Editor-in-Chief

PLOS Pathogens

orcid.org/0000-0002-7699-2064

This report provides a comprehensive comparative analysis of the tissue tropism, histopathological manifestations, and transcriptional changes following infection of cynomolgus macaques with two distinct strains of Lassa fever virus. Both reviewers were impressed by the scope of the study and the quality of the findings and are enthusiastic about the report. The reviewers provided comments that should help to clarify and improve the manuscript, which will likely primarily encompass edits and additions to the text.

We apologize for the length of time that it has taken to provide this feedback, which was due to one reviewer failing to return their comments.

Reviewer Comments (if any, and for reference):

Reviewer's Responses to Questions

**Part I - Summary**

Reviewer #1: The purpose of this study was to expand upon the authors’ previous work on a new cynomolgous macaque model of Lassa Fever. The paper presents a thorough investigation into tissue tropism, viral dissemination, and survival outcomes in a non-human primate model infected with different LASV strains, AV or Josiah. The primary objective was to elucidate potential immunological differences between these strains and their implications for pathogenesis and survival. Notably, the Josiah strain exhibited increased viral dissemination and resulted in a heightened host inflammatory response compared to AV, despite identical infection doses and routes. Furthermore, Josiah strain infection proved fatal in the non-human primate model, whereas AV-infected monkeys survived. The study's transcriptomic analysis was extensive, providing datasets from multiple tissues (lymph nodes, spleen, PBMCs, liver, lung, kidney, and adrenal glands) and offering comparative insights between groups (AV vs. Josiah, Josiah vs. Mock, and AV vs. Mock). The analysis highlighted gene expression related to immune responses and tissue function in liver, lung, and kidney. Methods were thorough and would allow for reproduction of the experiments. Transcriptomic analysis was outlined along with github links to share their lab’s analysis pipeline. Additionally, histological examinations involved staining for various immune cell types over the course of multiple timepoints. Overall, this study provides valuable information on the tissues to which LASV can spread during the infection, the immune responses that may be induced, and how this can differ greatly based on the LASV strain used.

Reviewer #2: This paper by Hortion and colleagues describes studies aimed at a detailed characterization of the virological and host response parameters, as well as histopathological manifestations, associated with survival or death from Lassa virus (LASV) infection in a cynomolgus macaque (CM) model of Lassa fever (LF) disease using of two different LASV strains from lineage IV, AV and Josiah, which cause non-lethal and lethal, respectively, disease in CM.

The authors have conducted a very comprehensive analysis of virus propagation and tissue distribution over time, as well as the host immune responses including transcriptome responses in main organs targeted by LASV, and the characterization, based on deconvolution analysis of transcriptome data, of immune cell infiltrates.

The experimental component of the paper does not directly address mechanistic aspects of LASV pathogenesis but provides a wealth of high-quality results on both viral and host response parameters during infection with two related strains of LASV associated with very different outcomes in terms of disease severity. The information provided in the present work represents a highly valuable resource that provides the bases for future studies aimed at elucidating the viral and host factors, as well as mechanisms underlying the molecular and cellular bases of LASV pathogenesis.

The authors have provided convincing evidence that draining lymph nodes are the initial viral target, with infected cells being detected at 2 days post-infection in the subcortical sinus of LNs close to the inoculation site. Subsequently, the virus is detected in different LNs, including GALT, thymus and spleen, supporting that secondary lymphoid organs are an early major location of virus multiplication for both highly virulent (Josiah) and mild disease causing (AV) LASV strains in CM. However, viral tropism of these two LASV strains in LN showed some differences, with AV antigens found only in APCs, whereas Josiah antigen was detected in additional cell types.

After day 5 post-infection Josiah and AV strains start to exhibit very different patterns of spread. Josiah strain disseminated to almost all the organs, replicating in non-immune cells, stromal cells, endothelial cells, hepatocytes, and epithelial cells, whereas spread of the AV strain spread was restricted to liver, adrenal glands and capillaries in the brain. The pathological changes observed in the brains of Josiah-infected CM are consistent with encephalitis, which can account for neurological signs observed in most Josiah-infected cynomolgus monkeys.

The authors used a combination of IHC, ISH, transcriptome deconvolution and RNAseq of specific cell populations isolated from FFPE sections collected by laser-capture microdissection to generate a comprehensive analysis of the cell content in LNs and the transcriptome responses in lung, kidney, spleen, liver, and adrenal gland tissues during infection with the two LASV strains.

These studies revealed that fatal infection with Josiah strain was associated with immune cell infiltrates in most tissues.

The author’s finding of immune cell infiltrates including CD4+ T cells, NK cells, mDCs, and monocyte in the lungs, as well as high levels of mRNAs related to the type I IFN response, monocyte and T-cell activation, cytokines and chemokines, together with the thickening of the alveolar walls can account for the acute respiratory distress syndrome observed in fatal LF.

The authors documented signatures of the activation of type I IFN response, cytokines and chemokines, and coagulation pathways in both infected and bystander hepatocytes in both Josiah-and AV-infected CM, despite AV not being detected in liver, suggesting that liver inflammation was driven by the host inflammatory response rather than virus multiplication in liver. The authors documented a similar pattern in the kidneys of CM infected with both Josiah and AV strains, indicating that this activation was not directly caused by virus targeting the kidney as no kidney infection was detected AV-infected CM.

The results presented by the authors confirmed the adrenal gland as a main site of Josiah multiplication, which may affect the renin- angiotensin-aldosterone system that plays a key role in controlling blood pressure, fluid balance and ionic composition. The increase in renin concentration in the blood of Josiah-, but not AV-infected animals is consistent with the pathogenicity of Josiah infection. High aldosterone levels observed at peak disease in fatal LF were probably induced in response to renin stimulation and therefore represent the physiological response induced to restore blood pressure, whereas increased cortisol levels in Josiah-infected CM probably reflects a stress response to infection. These findings suggest that despite robust viral replication in the adrenal glands, the endocrine functions of these organs remain largely unaffected. The authors also documented the detection of LASV RNA in the testes of surviving animals one month after infection, a finding consistent with the recent demonstration of long-term LASV persistence in testes raising the potential risk for sexual transmission mediated by individuals fully recovered from LF.

**Part II – Major Issues: Key Experiments Required for Acceptance**

Reviewer #1: 1. It’s important for the authors to comment on why they didn’t include a survival curve or tracking of clinical symptoms in each group of NHPs, and whether this could be added to help emphasize the stark differences between the infection groups (AV vs Josiah vs Mock).

2. The extensive histology and transcriptomic data are sufficient to back up their claims that secondary lymphoid organs are early and crucial reservoirs for LASV replication, and control or spread of the infection from the SLOs is potentially an indicator for recovery vs. catastrophic disease due to the LASV infection. However, it would be important to note what cell types were infected in the thymus and pancreas, etc. It would also be interesting to correlate viral infection and infiltration of neutrophils and macrophages (ISH) images with H&E changes—can H&E images be included?

Reviewer #2: The authors showed that Josiah, but not AV, strain has a very strong tropism for endothelial cells, which may contribute to its wide tissue/organ spread. Considering the apparent critical role played by the virus-endothelial cell interaction in viral dissemination, it would be of interest use cell-based assays to examine differences in endothelial susceptibility to Josiah and AV strains, as well as possible differences in changes in endothelial cell permeability following infection with Josiah or AV strains.

I concur with the authors that their findings support secondary lymphoid organs (SLOs) as the main early sites of LASV multiplication in CM infected with either the highly virulent Josiah strain, or the mild disease-causing AV strain. The findings presented in the paper also show that after 5 days post-infection, the cell tropism, viral dissemination, and immune cell infiltrattes strikingly differ between CM infected with Josiah and AV strains. However, no evidence has been presented that events that take place in SLOs early in the infection determine the outcome of LASV infection.

The authors have used a route of infection (SC) that may not accurately reflect some of the early events taken place during a more likely natural route of infection (in, aerosols). This issue should be discussed.

LASV AV strain was isolated from a fatal case of LF, whereas in CM, AV causes a mild disease. The authors should discuss these significant differences and possible limitations on the interpretation of their results regarding the events taken place in human cases of LF.

**Part III – Minor Issues: Editorial and Data Presentation Modifications**

Reviewer #1: 1. Figures could be clarified by adding consistent titles/headers, some suggestions:

a. Figure 1 - clearer labeling of which panels were 5dpi and 11dpi

b. Figure 5B - a header of “PBMCs” could be added like the headers in 5A

c. Figure 8C - quadrant depiction of which chart is which could be replaced with small headers over each chart

d. Figure 9B and 9C need labels for what each of the graphs are on the figure.

2. It might be important to discuss further the different disease outcomes mediated by two different strains of LASV vs. the potential host immune responses that might also be altering the disease outcome.

Reviewer #2: I would suggest to the authors trying to simplify the writing of the description of the events take place at different times post-infection and in different organs. I think that a cartoon showing viral distribution and main host responses over the course of infection will be helpful for the readers.

PLOS authors have the option to publish the peer review history of their article (what does this mean?). If published, this will include your full peer review and any attached files.

Reviewer #1: **Yes: **Hinh Ly

Reviewer #2: No

Figure Files:

Data Requirements:

Reproducibility:

References:

---

## [Editor Report · Decision Letter 1]

22 Nov 2024

Dear Dr. Baize,

We are pleased to inform you that your manuscript 'Fatal Lassa fever in cynomolgus monkeys is associated with systemic viral dissemination and inflammation' has been provisionally accepted for publication in PLOS Pathogens.

Best regards,

Allan J Zajac

Academic Editor

PLOS Pathogens

Thomas Hoenen

Section Editor

PLOS Pathogens

Michael Malim

Editor-in-Chief

PLOS Pathogens

orcid.org/0000-0002-7699-2064

This revised report continues to provide new details about the pathobiology and immune response to two distinct strains of Lassa fever virus following infection of macaques. The reviewers were enthusiastic about manuscript during the first round of reviews. In the revised report the authors have responded appropriately to the reviewers comments. The text has been revised and new data regarding the infection of the thymus and pancreas by the Josiah strain are now included. 
---

## [Editor Report · Acceptance letter]

3 Dec 2024

Dear Dr. Baize,

We are delighted to inform you that your manuscript, " Fatal Lassa fever in cynomolgus monkeys is associated with systemic viral dissemination and inflammation ," has been formally accepted for publication in PLOS Pathogens.

Best regards,

Sumita Bhaduri-McIntosh

Editor-in-Chief

PLOS Pathogens

orcid.org/0000-0003-2946-9497

Michael Malim

Editor-in-Chief

PLOS Pathogens

orcid.org/0000-0002-7699-2064